# Methane retrieval from MethaneAIR using the $CO_2$ Proxy Approach: A demonstration for the upcoming MethaneSAT mission

Christopher Chan Miller[1,2,3,4], Sébastien Roche[1,2,3], Jonas S. Wilzewski[1,2,*], Xiong Liu[2], Kelly Chance[2], Amir H. Souri[2], Eamon Conway[2,5], Bingkun Luo[2], Jenna Samra[2], Jacob Hawthorne[2], Kang Sun[6,7], Carly Staebell[6], Apisada Chulakadabba[1], Maryann Sargent[1], Joshua S. Benmergui[1,3], Jonathan E. Franklin[1], Bruce C. Daube[1], Yang Li[8], Joshua L. Laughner[9], Bianca C. Baier[10], Ritesh Gautam[3], Mark Omara[3], and Steven C. Wofsy[1]

[1]Harvard John A. Paulson School of Engineering and Applied Sciences, Harvard University, Cambridge, MA, USA
[2]Center for Astrophysics | Harvard & Smithsonian, Cambridge, MA
[3]Environmental Defense Fund, New York, NY
[4]Climate Change Research Centre, University of New South Wales, Kensington, NSW, Australia
[5]Kostas Research Institute for Homeland Security, Northeastern University, Burlington, MA, USA
[6]Department of Civil, Structural and Environmental Engineering, University at Buffalo, Buffalo, NY, USA
[7]Research and Education in Energy, Environment and Water Institute, University at Buffalo, Buffalo, NY, USA
[8]Department of Environmental Science, Baylor University, Waco, TX
[9]Jet Propulsion Laboratory, California Institute of Technology, Pasadena, CA, USA
[10]NOAA Global Monitoring Laboratory, Boulder, CO, USA
*Now at: EUMETSAT, Eumetsat Allee 1, 64295 Darmstadt, Germany

**Correspondence:** Christopher Chan Miller (cmiller@g.harvard.edu)

**Abstract.**

Reducing methane ($CH_4$) emissions from the oil and gas (O&G) sector is key to mitigating climate change in the near-term. MethaneSAT is an upcoming satellite mission designed to monitor basin-wide O&G emissions globally, providing estimates of emission rates and helping identify the underlying processes leading to methane release to the atmosphere. MethaneSAT data will help advocacy and policy efforts to help track methane reduction commitments and targets made by countries and industry. Here we introduce the $CH_4$ retrieval algorithm for MethaneSAT based on the $CO_2$ proxy method. We apply the algorithm to observations from the maiden campaign of MethaneAIR, an airborne precursor to the satellite with similar instrument specifications. The campaign was conducted during winter 2019 and summer 2021 over three major US oil and gas basins.

Analysis of the MethaneAIR data shows that measurement precision is typically better than 2% for $20 \times 20$ m$^2$ pixel resolution, with no strong dependence on geophysical variables such as surface reflectance. We show that detector focus drifts over the course of each flight likely due to thermal gradients that develop across the optical bench. The impacts of this drift on retrieved $CH_4$ can mostly be mitigated by including a parameter that squeezes the laboratory tabulated instrument spectral response function in the spectral fit. Validation against coincident EM27/SUN retrievals shows that MethaneAIR values are generally within 1%. MethaneAIR retrievals were also intercompared with those of TROPOMI; The mean bias between instruments is 2.5 ppb, and the latitudinal gradients for the two datasets are in good agreement.

We evaluate the accuracy of MethaneAIR estimates of point source emissions using observations made over the Permian O&G basin, based on the integrated mass enhancement approach coupled with a plume-masking algorithm based on total variational denoising. We estimate that the median point source detection threshold is 100-150 kg h$^{-1}$ at the aircraft's nominal 12km above-surface observation altitude, based on an ensemble WRF large eddy simulations used to mimic the campaign conditions with the threshold for quantification about $2\times$ the detection threshold. Retrievals from repeated basin surveys indicate the presence of both persistent and intermittent sources, and we highlight an example from each case. For the persistent source we infer emissions from a large O&G processing facility, and estimate a leak rate between 1.6 and 2.1 %, higher than any previously-reported emission from a facility of its size. We also identify a ruptured pipeline that alone would constitute 2 % of estimated basin emissions, two weeks before it was found by its operator, highlighting the importance of regular monitoring from the future satellite mission. The results showcase the capability of MethaneAIR to make highly accurate, precise measurements of methane dry-air mole fractions in the atmosphere, with fine spatial resolution ($\sim 20 \times 20$ m$^2$) mapped over large swaths ($\sim 100 \times 100$ km$^2$) in a single flight. The results provide confidence that MethaneSAT can make such measurements at unprecedentedly fine scales from space ($\sim 130 \times 400$ m$^2$ pixel size over $\sim 200 \times 200$ km$^2$ target area), thereby delivering quantitative data on basin-wide methane emissions.

## 1  Introduction

Methane (CH$_4$) is the second most important human-influenced greenhouse gas (GHG), with a radiative forcing one third the magnitude of carbon dioxide (CO$_2$) (Etminan et al., 2016). Recently there has been considerable policy focus on reducing anthropogenic methane emissions, culminating in over 100 nations agreeing during the COP26 meeting in Glasgow in 2021 to a 30 % reduction of 2020 levels by 2030 (Malley et al., 2023). These reductions are expected to be driven in large part by tightened emission controls on the O&G sector (White House Climate Policy Office, 2021), owing mostly to their cost-effectiveness relative to other major anthropogenic sources (UNEP, 2021). Improved monitoring is required to help companies and regulators understand where and why methane emissions occur, to quantify emission rates in large oil and gas production regions, and to ensure that both nations and major O&G producers meet their stated commitments.

MethaneSAT, slated to be launched in early 2024, is a satellite mission designed with the primary goal of quantifying all CH$_4$ emissions from major O&G production basins at high spatial resolution with regular revisits. It is funded by private philanthropy and is managed by MethaneSAT LLC, a wholly-owned subsidiary of the Environmental Defense Fund. In preparation for MethaneSAT's launch, an airborne precursor called MethaneAIR has been constructed (Staebell et al., 2021), with near-identical instrument specifications (Table S1, Chulakadaba et al. (2023)). Here we use MethaneAIR observations from its maiden flight campaign to validate the MethaneSAT CO$_2$-proxy CH$_4$ algorithm used to retrieve dry-air column averaged CH$_4$ mole-fractions ($XCH_4$ ), and report on the accuracy of emission rate determinations using this sensor.

Remote-sensed CH$_4$ observations are the most efficient way for mapping methane emissions at large scale. Satellites have been monitoring CH$_4$ globally since the launch of SCIAMACHY in 2002 (Frankenberg et al., 2006). TROPOMI, the most recent of this class of global CH$_4$ mappers, was launched in 2017. It has $5.5 \times 7$ km$^2$ spatial resolution with daily global coverage,

producing observations with pixels $\sim 90\times$ smaller than SCIAMACHY, and covering the globe $6\times$ faster (Hu et al., 2018). This class of sensors, in particular GOSAT (Parker et al., 2020), have been used to regionally constrain methane emissions (Monteil et al., 2013; Turner et al., 2015; Maasakkers et al., 2019; Lu et al., 2021; Zhang et al., 2021; Qu et al., 2022), and thus proven useful for identifying drivers of global $CH_4$ trends (Deng et al., 2022; Janardanan et al., 2020; Worden et al., 2022). Currently TROPOMI has been less used in global inversions due to region-specific retrieval biases (Qu et al., 2021; Jacob et al., 2022), but its high spatiotemporal coverage makes it the first global mapper capable of constraining total O&G basin $CH_4$ emissions, and its retrievals have been used to reveal large emissions underestimates for multiple basins (Schneising et al., 2020; Zhang et al., 2020; Shen et al., 2021, 2022). However, reliable basin-wide emissions estimates require months of TROPOMI data due to the low 3% retrieval success rate, a result of unfavorable scene conditions (reflectance/cloud/aerosol), and the limited robustness of the full-physics approach (Jacob et al., 2022). The $CO_2$-proxy approach has proven more robust under moderate aerosol conditions (Parker et al., 2015), with the GOSAT $CO_2$-Proxy retrieval showing a 24% success rate (Parker et al., 2020). Here aerosol scattering is implicitly accounted for by normalizing the retrieved $CH_4$ column against a $CO_2$ column retrieved from the same spectral region. The $CO_2$-Proxy approach is not possible with TROPOMI as there is no nearby $CO_2$ absorption in the targeted 2.3 $\mu$m $CH_4$ band.

Recently instruments designed to detect high concentrations of $CH_4$ in individual methane plumes have been deployed on aircraft (AVIRIS-Thorpe et al. (2012), AVIRIS-NG-Thorpe et al. (2016), HySpex-Hochstaffl et al. (2023)) and satellites (Sentinel 2-Varon et al. (2021), GHGSat-Jervis et al. (2021), CarbonMapper-Shivers et al. (2021), PRISMA-Guanter et al. (2021), EnMAP-Roger et al. (2024)) to estimate emission rates from point sources. Such instruments have increased pixel resolution (O(1-10 m) length scale) at the expense of spectral resolution (O(10 nm) vs O(0.1 nm) for global mappers). At these spatial scales $CH_4$ concentrations from plumes originating from point sources can be much higher than the atmospheric background, loosening the retrieval accuracy requirement. They have proven particularly useful for monitoring emissions from O&G infrastructure where emissions from individual facilities follow heavy-tailed distributions (Brandt et al., 2016; Zavala-Araiza et al., 2015; Frankenberg et al., 2016; Cusworth et al., 2021), however it is not possible to determine what fraction of emissions within a basin are measured by observing a small number of detectable superemitters, and such measurements can miss a substantial proportion of emissions from smaller sources.

In fact, recent work has suggested these smaller sources represent a significant fraction of total O&G methane emissions. From a statistical survey of observed site-level O&G well emissions, Omara et al. (2022) estimated that low-production wells contribute over 50% of all US oil and gas production-related methane emissions, due to their high production-normalized leak-rates ($> 10\%$ on average) and prevalence (81% of all producing wells). However because emissions from individual low-producing wells are small (95% of sites emit less than 7 kg/h), they will be invisible to current and future satellite methane point-source instruments, which have detection limits at least an order of magnitude larger [*]. Aircraft sensors such as Airborne Visible InfraRed Imaging Spectrometer - Next Generation (AVIRIS-NG) have plume detection limits down to 10 kg/h (Thorpe et al., 2016), enabling detection of the highest-emitting low-production wells. However these reported detection limits correspond

---

[*]Carbon Mapper has the best reported point source detection limit (50-150 kg/h) of any current and future planned satellite methane point-source observer (Carbon Mapper, Inc., 2023)

to ground pixel sizes of $\sim 0.5$m, requiring observations to be made close to the surface (0.5-1km). To map a basin in such a manner would require many days of flying. Recently Cusworth et al. (2022) conducted a large survey of US O&G basins with the AVIRIS-NG instrument, flying at flight altitudes more reasonable for large-scale surveying (3-5 km). Through comparisons

with basin emissions inferred from TROPOMI, they estimate they were able to constrain about 35% of total emissions from oil and gas infrastructure, missing about two thirds of basin emissions. The ability of these sensors to assign sources to specific infrastructure provides some insight into the specific practices and activities contributing to the basin total emissions by large point sources. However, a complete monitoring strategy requires this type of sensor to be used in combination with sensors capable of regularly measuring total basin emissions and the spatial distribution of diffuse emissions.

MethaneSAT has been designed to fill the gap between space-borne global flux mappers and point-source imaging instruments. It contains a pair of 2-D grating spectrometers with a similar spectral resolution to TROPOMI, covering the 1.27 $\mu$m $O_2$ singlet delta ($a^1\Delta_g \leftarrow X^3\Sigma_g^-$) and 1.65 $\mu$m ($2\nu_3$) $CH_4$ bands. Rather than acquiring a broad swath to achieve daily global coverage like TROPOMI and upcoming missions such as CO2M (Sierk et al., 2019), it is maneuverable and built to target scenes at the approximate scale of an O&G production basin ($200 \times 200$ km$^2$ at nadir, for a 30 s collect). By concentrating

pixels onto the smaller acquisition area it achieves a pixel size of $130 \times 400$ m$^2$ in its nominal operational mode, enabling $CH_4$ to be retrieved at the accuracy of a TROPOMI like instrument (Jacob et al., 2022), but with a spatial resolution capable of imaging individual plumes from point sources. The unique combination of high spatial resolution, measurement precision, and a wide swath will enable both the estimation of point-source emissions from high emitting facilities, whilst also quantifying and mapping the entire basin source via the inversion of an emissions field, representing the sum of sources individually below the

plume detection threshold. MethaneSAT will acquire an average of 30 scenes per day enabling at least 10-20 revisits to major O&G basins per year. Targets will be prioritized based on production rates of oil and gas, and scheduled based on favorable meteorological forecast conditions (low cloud cover, steady winds) for both the retrieval and emissions inversion, maximizing the utility of each acquisition (Benmergui, 2019).

Here we present results from the maiden flight campaign of MethaneAIR using the operational MethaneSAT $CO_2$-proxy

$XCH_4$ retrieval. This is expected to be the primary $XCH_4$ product used in subsequent emissions inversions [†]. The $CO_2$-Proxy method was first used from an airborne platform by the MAMAP instrument (Krings et al., 2011; Gerilowski et al., 2011), which provided the first remote-sensed estimates of $CH_4$ and $CO_2$ point/small-area sources (Krautwurst et al., 2017; Krings et al., 2018). MethaneAIR builds on this heritage by substantially increasing the sensors spatial coverage rate, mapping approximately $490\times$ the area per unit time relative to MAMAP: This is achieved by its higher nominal operating altitude (12

110 km vs 1.25 km), faster aircraft speed/exposure time (720 vs. 200 km h$^{-1}$/ 0.1s vs. 0.6s), and its configuration as a push-broom scanner (983 vs. 1 across-track pixels). This allows complete mapping of a typical sized oil and gas basin in a few hours of flight time.

The road map for the rest of the paper is as follows; Section 2 describes the flight campaign, and Section 3 the retrieval methodology. Section 4 describes the method used for retrieval bias correction, needed due to drifts caused by impact of cabin

temperature changes during flight. The MethaneAIR observations are validated against ground-based (EM27/Sun) and satellite

---

[†]We are also exploring alternate approaches for targets containing expected $CO_2$ enhancements (Section S2.2)

(TROPOMI) retrievals in Section 5. The results of this validation are used to estimate the MethaneAIR detection limit for point sources, and further discuss challenges for the $CH_4$ emissions inversion problem at fine spatial scales based on the observations (Section 6). We highlight some case-studies from observations in the Permian in Section 7. Lastly, in Section 8 we discuss the implications for MethaneSAT.

## 2    First Flight Campaign

The first two MethaneAIR flights took place in November 2019 on board the NSF/NCAR High-performance Instrumented Airborne Platform for Environmental Research (HIAPER) GV aircraft (UCAR/NCAR-Earth Observing Laboratory, 2005). The remaining 8 flights were performed during July-August 2021, delayed initially by aircraft issues, and later by the COVID-19 pandemic. Figure 1 shows the flight path for the first 9 flights, where the instrument was operating in nadir viewing mode. For the final flight the instrument orientation was flipped to zenith orientation to observe oxygen airglow in the early evening.

The first 3 flights (RF01-RF03) were performed for engineering assessment and instrument function. These were conducted around the Colorado front range region near the project base airport in Broomfield CO, and were intended to avoid major $CH_4$ emission sources to provide background conditions for evaluating instrument performance. In RF02 a linear-polarizer was placed in front of the instrument at 3 different angles to test polarization sensitivity. However it also induced an additional defocusing effect from lensing making evaluation of the impact of polarization difficult. The remaining 6 nadir viewing flights focused on mapping $XCH_4$ in the Permian Basin (RF04-RF07), Uintah Basin/ Salt Lake City (RF08) and Bakken Formation (RF09). In RF04 and RF05, the plane also made several overpasses over a controlled $CH_4$ release in Midland TX.

$CH_4$ and $CO_2$ mole fractions were retrieved for validation using EM27/SUN FTIR spectrometers for the duration of the campaigns (Gisi et al., 2012; Frey et al., 2019; Alberti et al., 2022). In 2019 two instruments (instrument IDs HA, HC) were concurrently operated 1 km apart east of Fort Collins, CO (40.809°N 104.777°W and 40.806°N 104.756°W respectively), to evaluate the observing system's ability to measure $XCH_4$ gradients. In 2021 the observations were performed from a single instrument located on the roof of the NOAA ESRL building in Boulder CO (instrument ID KB, 39.991°N 105.261°W). All EM27/SUN retrievals shown here were processed using the latest version of the TCCON retrieval code (GGG2020, Laughner et al., 2023b). In-situ $CH_4$ and $CO_2$ were made on the aircraft using a Picarro G2401-mc cavity ring-down spectrometer. The aircraft performed a series of missed runway approaches to enable the in-situ data to be used to evaluate the *a priori* gas profiles.

## 3    Methane Retrieval

The MethaneAIR instrument consists of two Offner spectrometers (Headwall Photonics), covering 1237–1319 nm and 1592–1697 nm wavelength ranges, recorded with InGaAs detectors (Princeton IR Technologies). For this paper we focus on the longer wavelength spectrometer, which records gas absorption from both the P and R branches of the 1.6 $\mu$m $CO_2$ band, and the $2\nu_3$ $CH_4$ band at $\sim$ 0.28 nm Full Width at Half Maximum (FWHM) resolution. This enables the use of the $CO_2$ proxy

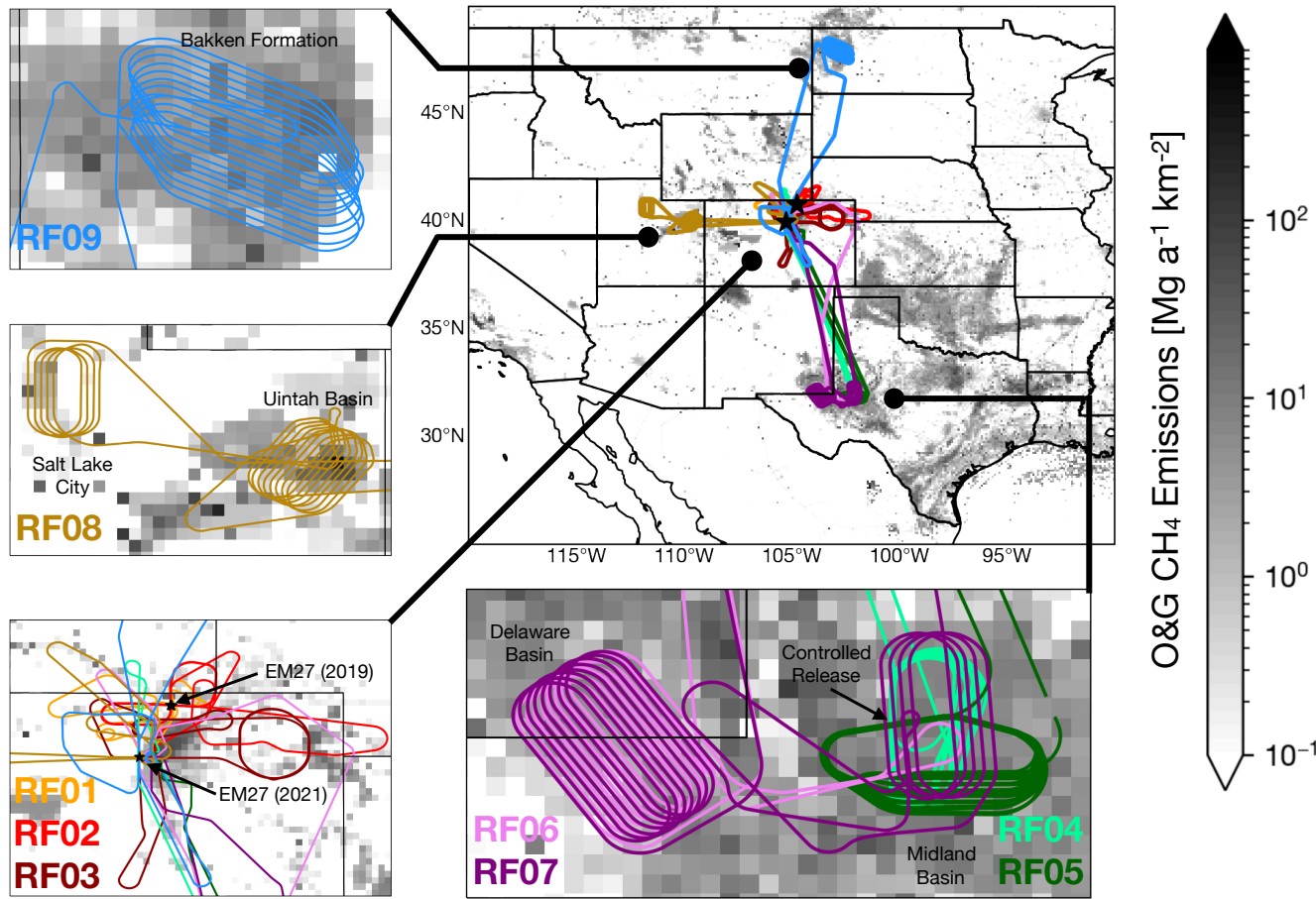

**Figure 1.** Aircraft flight paths for the 9 nadir-viewing research flights from the first campaign, overlaid against total oil and gas emissions (Scarpelli et al., 2020). Black stars indicate locations of the EM27/SUN spectrometers used for validation.

method (Frankenberg et al., 2005, 2006; Krings et al., 2011) for retrieving total column-averaged dry-air mole fractions of $CH_4$ ($XCH_4$). The instrument operates in pushbroom mode through an anti-reflective coated nadir port in the aircraft, with a focal plane array (FPA) of 1024 spectral $\times$ 1280 spatial pixels. In practice the output dimensions of MethaneAIR data products are 1024 $\times$ 983, because the projected slit image does not fully illuminate the entire FPA cross-track width. At the nominal 12 km flight altitude, the swath width is $\sim$ 5 km. For the majority of results presented here we aggregate the cross-track pixels by a factor of 5 for computation expediency and to increase the signal-to-noise ratio, yielding a ground pixel size of $\sim 20 \times 20$ m$^2$. A more detailed description of the instrument, aircraft integration, and calibration is presented in Staebell et al. (2021). The operational Level 0–1 processor to produce radiometrically calibrated and geolocated radiance is described in Conway et al. (2023).

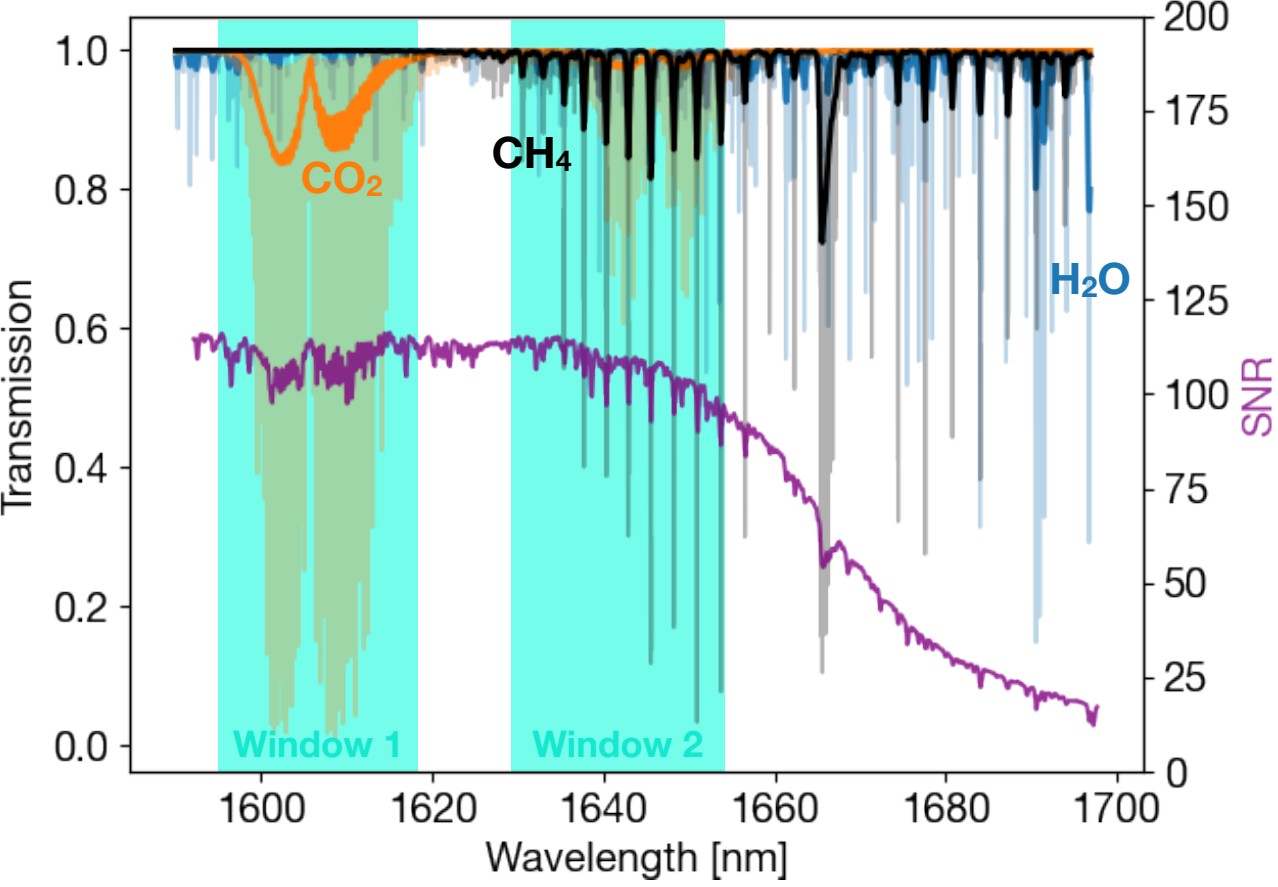

**Figure 2.** Example transmission spectra for a typical MethaneAIR observation (0.3 Lambertian albedo, $30°$ solar zenith angle). Transparent and solid lines show spectra before and after convolution with the MethaneAIR instrument line shape respectively. The signal-to-noise ratio for a typical measurement is also indicated. Windows 1 and 2 indicate the spectral regions to retrieve $CO_2$ and $CH_4$ respectively.

$XCH_4$ is retrieved using the Smithsonian PLanetary ATmosphere retrieval (SPLAT), a new flexible optimal-estimation retrieval developed at the Harvard-Smithsonian Center for Astrophysics for use in both Earth and other planetary atmosphere inverse problems. Here we use SPLAT to infer the $CH_4$ and $CO_2$ vertical column densities ($N_{CH_4}$ and $N_{CO_2}$ respectively). The proxy-derived $XCH_4$ is

160 $$XCH_4 = \frac{N_{CH_4}}{N_{CO_2}} XCO_{2,0} \tag{1}$$

where $XCO_{2,0}$ is an *a priori* estimate of the column averaged dry-air $CO_2$ mole fraction. To derive $N_{CO_2}$ and $N_{CH_4}$ we target wavelength windows 1595–1618 nm and 1629–1654 nm respectively (Figure 2). Although MethaneSAT can only observe the part of the R branch above 1597 nm, here we leverage MethaneAIR's broader spectral window and include data up to 1595 nm to increase the retrieval $CO_2$ precision. For MethaneSAT, the cost of not observing the R branch will be partially

offset by its higher spectral resolution, with a lineshape FWHM $\sim 30$ % narrower. An additional set of weaker $CO_2$ lines overlap with the $CH_4$ R-branch, potentially providing a better light path constraint, but are likely too weak to provide a good retrieval of $N_{CO_2}$ alone. For the $CH_4$ window, the 1654 nm upper limit has been constrained by the InGaAs detector quantum efficiency, which begins rolling off from 70 % below 1650 nm to 26 % by 1670 nm. The mercury–cadmium– telluride (MCT) detectors used in MethaneSAT do not show this roll off, which may enable a wider fit window.

## 3.1 Spectroscopy

We model $CH_4$, $CO_2$, and $H_2O$ absorption using cross section lookup tables computed from the GGG2020 spectral database used operationally in TCCON retrievals (Wunch et al., 2011a; Toon, 2022b, a). These include speed-dependent Voigt line shapes with line mixing for $CO_2$ (Mendonca et al., 2016) and $CH_4$ (Mendonca et al., 2017). We opt to use GGG2020 to leverage the previous validation work evaluating the TCCON retrievals against profiles constructed from in-situ aircraft and

175 balloon observations (Wunch et al., 2010; Messerschmidt et al., 2011; Geibel et al., 2012). We scale the retrieved $XCH_4$ by the ratio of the TCCON GGG2020 airmass-independent correction factors for CO2 and CH4 (1.0101/1.0031). This is intended to account for the bias induced by the collective effect of GGG2020 line strength errors across the $CO_2$ and $CH_4$ bands. In actuality the factor for MethaneAIR may differ slightly since there are vertical sensitivity differences between it and the TCCON retrievals from which the correction factors are derived. We plan to revisit the question of absolute scaling when we

have accumulated more ground-validation overpasses in future campaigns.

## 3.2 Retrieval Configuration

$N_{CO_2}$ and $N_{CH_4}$ are derived through optimization of a state vector ($\mathbf{x}$) that minimizes the following cost function $J(\mathbf{x})$.

$$J(\mathbf{x}) = (\mathbf{y} - \mathbf{F}(\mathbf{x}))^T \mathbf{S_o}^{-1}(\mathbf{y} - \mathbf{F}(\mathbf{x})) + \gamma^{-2}(\mathbf{x} - \mathbf{x_a})^T \mathbf{S_a}^{-1}(\mathbf{x} - \mathbf{x_a}) \tag{2}$$

The above minimizes a balance between the fit residuals between the observations $\mathbf{y}$ and forward model $\mathbf{F}(\mathbf{x})$, and departure

of $\mathbf{x}$ from it's *a priori* estimate $\mathbf{x_a}$. The balance of both terms is controlled by the observation ($\mathbf{S_o}$) and *a priori* ($\mathbf{S_a}$) covariance matrices. $\gamma$ is an additional regularization parameter that scales the *a priori* covariance. For retrievals on the $5 \times 1$ aggregated pixels we select $\gamma^2 = 10$, guided by an L-curve analysis (Hansen, 1993), and the fact that it produces near-unity sensitivity to $CH_4$ in the boundary layer (see Section S1). Based on the same analysis we found the need to reduce the regularization for retrievals at the native pixel resolution ($\gamma^2 = 50$).

$\mathbf{y}$ contains radiances from both windows. The initial rationale for their joint optimization is that the weaker $CO_2$ band at 1.645 $\mu$m may improve the light path constraint due to its overlap with the target $CH_4$ band. In practice we have found little

difference compared to optimizing the windows independently. Since the proxy method accounts for aerosols via normalization against $N_{CO_2}$ as the light path constraint, $\mathbf{F}(\mathbf{x})$ does not consider scattering and is modeled analytically (Frankenberg et al., 2005). This allows for significantly faster processing compared to the "full-physics" approach, which explicitly includes

aerosols in the state vector. MethaneAIR generates approximately 30 million spectra per flight hour at native spatial resolution, making speed a significant consideration. The fastest full-physics $CH_4$ retrievals have processing times of $\sim 10$ s (Hu et al., 2018), whereas the proxy retrieval is an order of magnitude faster ($\sim 1$ s per pixel). Thus the $CO_2$ proxy approach will likely be the backbone for the future MethaneSAT operational processing, with a full-physics algorithm additionally applied at a aggregated resolution, or for selective cases where the *a priori* $CO_2$ is expected to be unreliable (e.g. for urban targets).

**Table 1.** MethaneAIR Level 2 algorithm fit settings

| State Vector Element | A Priori | Uncertainty ($1\sigma$) |
| --- | --- | --- |
| $CH_4$ Profile [19 Layers] | GGG2020 Priori Profile Software[1] | UoL GOSAT Proxy Covariance[2] |
| $CO_2$ Profile [19 Layers] | GGG2020 Priori Profile Software[1] | UoL GOSAT Proxy Covariance[2] |
| $H_2O$ Column | GEOS-FP[3] | 0.02 v/v |
| Temperature Profile Shift | GEOS-FP[3] | 5 K |
| Surface Pressure | GEOS-FP[3] | 4 hPa |
| Albedo[4] | MethaneAIR Radiance[6] | 100% |
| Radiance Offset[5] | 0.0 | $5 \times 10^{14}$ photons $cm^{-2}$ $s^{-1}$ $sr^{-1}$ $nm^{-1}$ |
| Wavelength Offset | 0.0 nm | 0.01 nm |
| ISRF squeeze | 1.0 | 0.2 |

[1] Laughner et al. (2022, 2023a)

[2] Covariance matrix from University of Leicester GOSAT Proxy retrieval (Parker et al., 2020)

[3] Rienecker et al. (2008)

[4] A third order Chebyshev polynomial is used to parameterize albedo for each window

[5] A first order Chebyshev polynomial is used to parameterize the radiance offset for each window

[6] Albedo estimated from 5-wavelength-pixel average centered at 1622.5 nm.

Table 1 summarizes the state vector used in the retrieval. A fuller description of how the state vector is implemented is provided in the supplement (Section S1). The settings are mostly consistent with similar GHG retrievals (O'Dell et al., 2012; Schepers et al., 2012; Parker et al., 2020). $CH_4$ and $CO_2$ profiles are optimized on a 19 layer vertical grid, consisting of 13 evenly spaced pressure layers from the surface to the tropopause, with a set of fixed pressure levels above. We also tested scaling the $CH_4$ and $CO_2$ columns, however this tends to overestimate column $XCH_4$ for cases with large surface enhancements

because absorption near the surface is more efficient due to increased pressure broadening encroaching on more transparent regions of the spectrum. Scaling the column artificially adds more $CH_4$ to higher altitudes where $CH_4$ absorbs mostly in the saturated part of the line, which requires more $CH_4$ to be added to fit the observed radiance over regions with surface enhancements.

Initial analysis of the first MethaneAIR flight data made apparent that the instrument spectral response function (ISRF) was drifting during the course of each flight. The instrument's low F-number (3.5) means that small perturbations to the instrument, such as changes in spectrometer optical bench temperature or the mechanical stresses at the interface between the spectrometer and camera can lead to significant impacts on light focus on the detector. In order to account for this, two additional ISRF squeeze parameters ($x_{sqz}$) have been included in the state vector to model the ISRF change over both fit windows. The changing width of the ISRF is modeled by squeezing the original wavelength grid from the laboratory-derived tabulated ISRF ($\Gamma_{TAB}(\lambda)$)

$$\Gamma(\lambda) = \Gamma_{TAB}(x_{sqz}\lambda) \tag{3}$$

From Equation 3, $x_{sqz}$ values below/above unity correspond to stretching/squeezing the tabulated ISRF respectively. Figure 3 shows the impact of including the ISRF correction on the spectral fit over a flat background region from the plane's return transit to Colorado from the Midland basin in RF05. When the ISRF squeeze parameter is not included, the fit residuals are significantly larger than expected from the laboratory-derived ISRF (Figure 3, middle panel). Including the ISRF squeeze brings the residuals within the expected noise level, and leads to a significant reduction in the $XCH_4$ cross-track bias. In this case the retrieved scaling factors ($x_{sqz}$) show up to 30 % changes in the ISRF width compared to laboratory calibrations. Improvements to the thermal housing of the instrument are planned to improve its stability in upcoming campaigns.

### 3.3 A Priori State

The Goddard Earth Observing System - Forward Processing (GEOS-FP) reanalysis (Rienecker et al., 2008) is used as the primary dataset for the construction of the prior. Profiles of pressure, temperature, and water vapor ($H_2O$) are sampled directly from GEOS-FP. Model-observation height differences computed using digital elevation tiles from Amazon Web Services (Larrick et al., 2020) are used to adjust the GEOS-FP surface pressures to the MethaneAIR ground pixel locations. $CO_2$ and $CH_4$ *a priori* profiles are calculated using the TCCON GGG2020 profile construction tool (Laughner et al., 2022), using GEOS-FP meteorology as inputs [‡]. The *a priori* Lambertian surface albedo for each pixel is computed using the transparent region of the observed radiance at 1622 nm, assuming a non-scattering atmosphere. The *a priori* uncertainties for most state vector elements are based on OCO-2 ACOS algorithm (O'Dell et al., 2012). The profile $CH_4$ and $CO_2$ covariance matrices are from the University of Leicester (UoL) GOSAT Proxy retrieval (Parker et al., 2020).

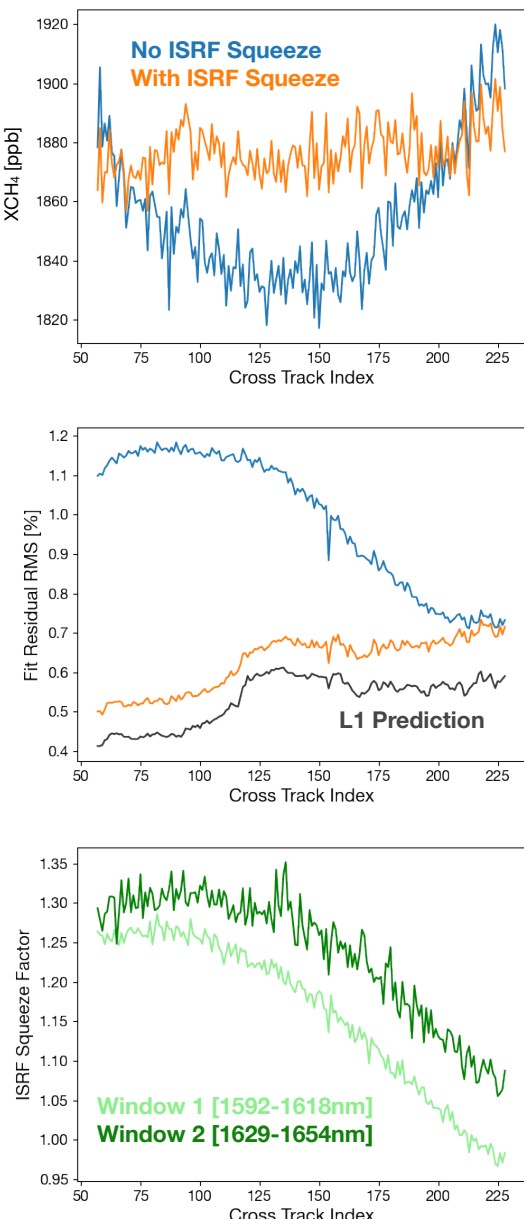

**Figure 3.** Impact of ISRF squeeze on retrieved $CH_4$, using observations over a clear region in RF05. Top: Cross-track averaged retrieved $XCH_4$ for retrievals with and without squeeze factor. Middle: Corresponding fit residual root mean square errors. "L1 Prediction" corresponds to the residual RMS expected from the radiance uncertainty in the L1 product. Bottom: Retrieved squeeze factors for the $CO_2$ (1595–1618 nm) and $CH_4$ (1629–1654 nm) windows.

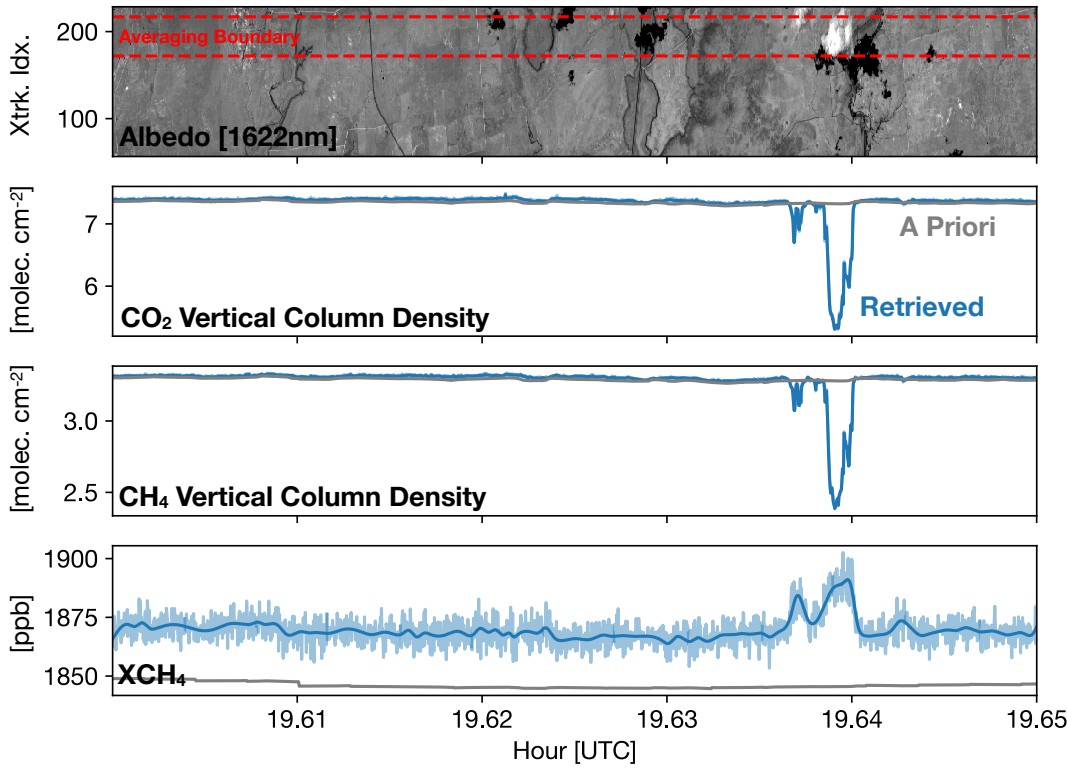

**Figure 4.** $XCH_4$ retrieved in the presence of a low-altitude cloud during RF06. The top plot shows the grayscale albedo image estimated from the MethaneAIR radiance. The next three plots from top to bottom show the cross-track median retrieved $CO_2$ vertical column, $CH_4$ vertical column, and $XCH_4$, computed within the averaging boundary indicated on the top panel. The solid blue lines are the 5-fold cross validated smoothed spline values. The *a priori* values are also shown in gray.

## 3.4 Cloud Screening

Cloud-impacted observations have been rejected using a cloud screening algorithm modeled on the $OCO_2$ A-Band Preprocessor (Taylor et al., 2016). Here surface pressure is retrieved from the instrument's $O_2$ band assuming a cloud-free atmosphere. In this case large deviations from the *a priori* pressure can be interpreted as due to clouds. The algorithm similarly uses retrieved $CO_2$ and $CH_4$ vertical column densities to screen out clouds with high optical depths, which cause distinct decreases relative to their priors (Figure 4). The screening flags are combined with the oxygen-band retrieval using a naive Bayes classifier (Heidinger

et al., 2012), which also enables the screening algorithm to work where there is no overlapping oxygen-band data. More details will be provided in a manuscript currently in preparation.

---

‡This will yield slightly different *a priori* profiles to the operational TCCON retrieval, which uses GEOS-FPIT, a frozen version of the GEOS-FP reanalysis

## 4 Cross-track bias correction for ISRF drift

### 4.1 Evidence of time-dependent cross-track bias

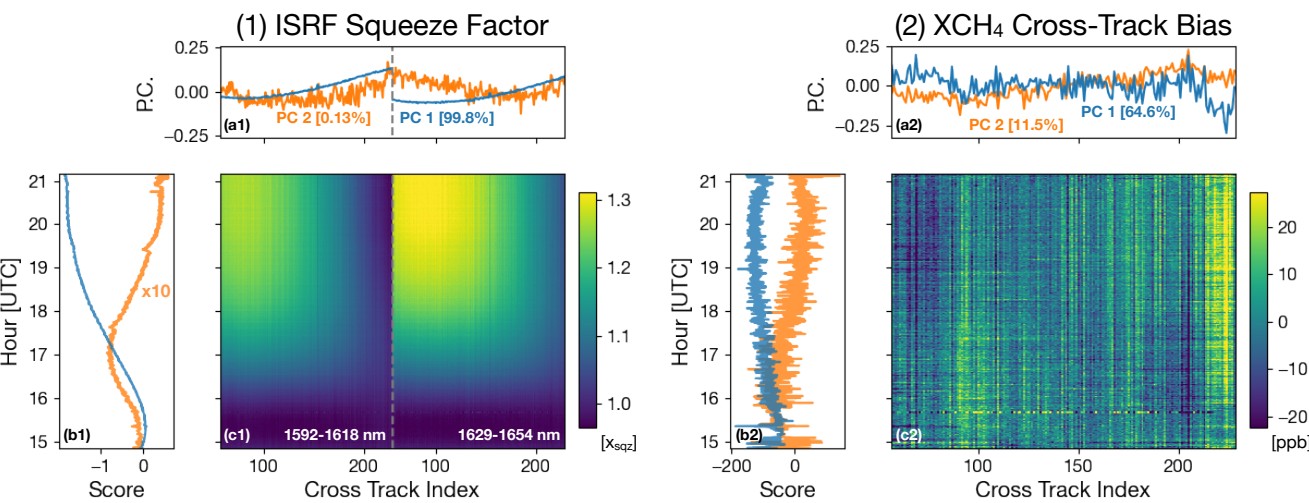

**Figure 5.** Relationship between retrieved ISRF squeeze factors and cross-track $XCH_4$ bias during RF05 (3rd August 2021). The contour plots show the time evolution of the ISRF squeeze factors for each retrieval window (c1) and $XCH_4$ cross-track bias derived from the small area approximation (c2). Top panels (a1) and (a2) show the first two principal components for the combined ISRF squeeze parameters and $XCH_4$ bias respectively, and the corresponding component scores are shown in panels (b1) and (b2).

In the previous section it was shown that the in-flight ISRF can differ significantly from the table derived from on-ground laser calibration measurements, with up to 30 % changes in FWHM (Figure 3). This change slowly evolves over the course of a flight. Figure 5(c1) shows the time-evolution of the ISRF squeeze factors for each spectral fit window retrieved from RF05. Early into the flight the ISRF squeeze factors ($x_{sqz}$) are close to 1, indicating the ISRF is close to the nominal calibration. As the flight continues, the ISRF width gradually narrows ($x_{sqz} > 1$), with the effect more pronounced at the side of the detector corresponding to the lower cross-track indices. Similar ISRF squeeze changes were observed during the other flights.

In order to better understand the temporal evolution of the ISRF, we performed a Principal Component Analysis (PCA) on the ISRF squeeze factors on both $CO_2$ and $CH_4$ fit windows simultaneously. PCA is a common dimensionality reduction technique that reconstructs a multidimensional dataset from a smaller number of principal components. Let $\mathbf{s}(t_i) \in \mathbb{R}^{2n}$ represent the vector containing the ISRF squeeze factors from both windows, each with $n$ cross-track pixels, at the $t_i^{th}$ time. $\mathbf{s}(t_i)$ is reconstructed using the $n_{pc}$ principal components ($\mathbf{p}_j \in \mathbb{R}^{2n}$), scaled by the scores $c_j(t_i)$.

$$\mathbf{s}(t_i) = \bar{\mathbf{s}} + \sum_{j=1}^{n_{pc}} c_j(t_i)\mathbf{p}_j \tag{4}$$

$\bar{\mathbf{s}}$ is the mean ISRF squeeze at each cross-track pixel. The first principle component $\mathbf{p}_1$ is chosen to account for the largest possible variance in the dataset, and each succeeding component accounts for the highest possible variance under the constraint that it is orthogonal to the previous ones.

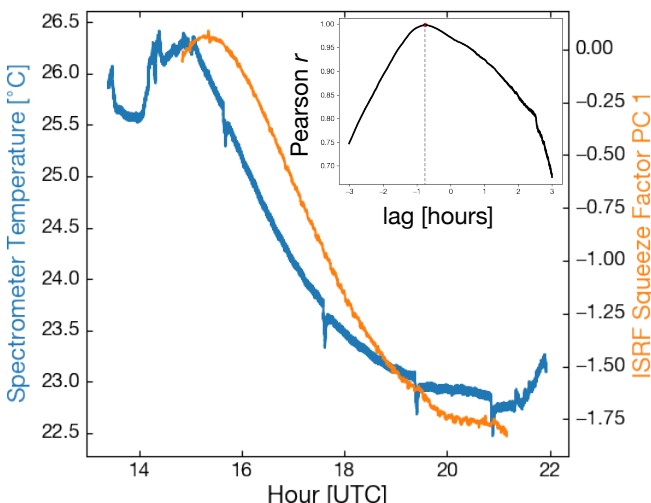

**Figure 6.** Comparison of the time-series of the first principal component of the ISRF squeeze factor (see text) and spectrometer temperature from RF05. Here spectrometer temperature refers to that recorded by a probe outside the instrument but within the thermal housing isolating the spectrometer from the plane cabin. The lag-correlation between the two variables is shown in the inset, defined as the Pearson $r$ value computed between the spectrometer temperature ($T_{spec}(t)$) and the ISRF principal component shifted by the lag value ($c_1(t + t_{lag})$)

.

Figure 5(a1) and (b1) shows the first two principal components and their scores respectively for the ISRF squeeze factors. The first principal component explains almost all the variance in the ISRF squeeze factors (99.8%). It is also highly correlated with the temperature of the environment surrounding the instrument (Figure 6). This strong relationship suggests that the ISRF changes are being caused by the cooling of critical spectrometer optical components, such as the foreoptic lens adjacent to the cold viewport window glass. These temperature changes can defocus light at the FPA leading to the observable changes in ISRF width. The fact that the ISRF changes lag the temperature by $\sim 0.75$ h also support this hypothesis (Figure 6 inset), as this would be expected due to the thermal inertia of the optical components.

It is unlikely that squeezing the tabulated laboratory ISRF fully accounts for the change in ISRF shape induced by defocusing. The gradual drift in instrument focus may lead to a time-dependent $XCH_4$ cross-track bias, which we attempt to derive using a "small area approximation" (O'Dell et al., 2018). This assumes $XCH_4$ over a small area is constant. In this case we derive the background $XCH_4$ for every cross-track by computing its median value over consecutive 10 s ($\sim 2$ km) intervals. The segment interval is chosen to be short enough so that the sequence of retrieved squeeze values captures temporal changes in the cross-track bias pattern, whilst long enough to reduce the impact of plumes, but not so long as to entrain topographic

gradients. Let $XCH_{4,g}(ix, it)$ be the retrieved $XCH_4$ for 10 s segment $g$ at cross/along track index $ix$ and $it$ respectively. The cross-track bias at cross-track $ix$ for the granule ($B_g(ix)$) is estimated as follows:

$$XCH_{4,g}^{med}(ix) = \text{med}\left(XCH_{4,g}(ix,:)\right) \tag{5}$$

$$B_g(ix) = XCH_{4,g}^{med}(ix) - \text{mean}\left(XCH_{4,g}^{med}(:)\right) \tag{6}$$

In the above med and mean denote the median and mean respectively, and : denotes the array axis in which the operation is taken over. The $XCH_4$ cross-track bias for RF05 is shown in Figure 5(c2), as well as its PCA decomposition (Figure 5(a2,b2)). The derived bias pattern shows a temporally constant set of index-to-index cross-track stripes over the course of the flight, with an additional slowly-evolving bias pattern smoothed over the entire detector array. The relatively constant pattern is likely due to instrument slit inhomogeneities, and is a common feature of other 2D grating spectrometers (e.g. MODIS: Rakwatin et al. (2007), OMI: Boersma et al. (2011), and TROPOMI: Borsdorff et al. (2019)).

The temporal evolution of the broader bias pattern strongly correlates to the change in ISRF width. This can be seen more clearly from the ISRF squeeze and $XCH_4$ cross-track bias patterns PCA scores (Figure 5(b1,b2)). The scores of the leading ISRF and $XCH_4$ bias PC are highly correlated in time (Pearson $r = 0.76$). Although the second ISRF PC explains very little of the total ISRF variability (0.13%), its scores are highly correlated with the first two $XCH_4$ bias PCs (Pearson $r$ values of 0.77 and 0.69 respectively). This suggests that subtle ISRF changes captured by the less-dominant PCs could contain valuable information for modeling the $XCH_4$ bias.

## 4.2 MethaneAIR cross-track bias correction algorithm

Since there is an underlying physical connection between the $XCH_4$ cross-track bias and ISRF squeeze parameters, errors associated with the small area approximation can be further reduced by constructing a regression model relating the two retrieved quantities. As the noise in retrieved ISRF squeeze parameters is lower than the retrieved $XCH_4$, this will also improve the precision of cross-track bias prediction compared to direct application of the values in Figure 5 (c2). Here we create a linear model of the cross-track bias ($n$ pixels) from a total number of $t$ segments, each containing 10 s of observations. We predict the $XCH_4$ biases (the "response" variables) derived from the small-area approximation $\mathbf{B} \in \mathbb{R}^{t \times n}$ against the retrieved ISRF squeeze factors (the "predictor" variables) combined from both windows $\mathbf{S} \in \mathbb{R}^{t \times 2n}$.

$$\mathbf{B} = \mathbf{S}\boldsymbol{\beta} \tag{7}$$

$\boldsymbol{\beta}$, the transformation between $\mathbf{S}$ and $\mathbf{B}$, is determined using partial least squares (PLS) regression (Wold et al., 2001). In this case an ordinary multiple least squares regression is not appropriate as it assumes there be no correlation between the squeeze factors at different cross-track positions. One possible way around this is to create a multiple regression model using a truncated set of principal components (e.g. those shown in Figure 5), but the principal components that are omitted from the regression could still contain valuable information for explaining variation *between* the response and predictor variables.

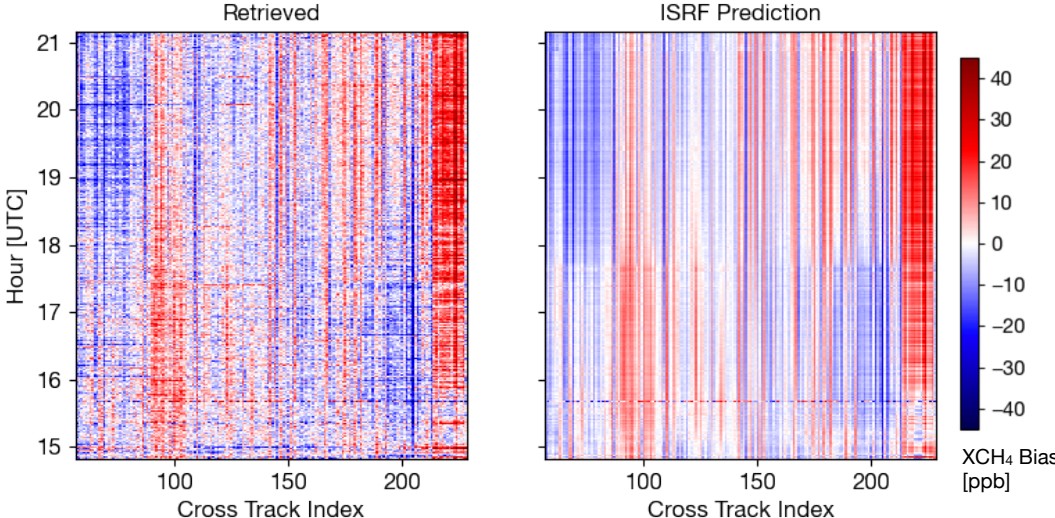

**Figure 7.** $XCH4$ cross-track bias estimate with the small area approximation (left) and subsequently refined using the PLS regression model (right). The PLS regression model (Equation 7) uses the retrieved ISRF squeeze factors to predict the data in the left panel, in order to preserve only sources of variability related to the temperature-induced defocusing effects.

Indeed, the strong correlation between the scores of the second ISRF PC with those from the $XCH_4$ bias dataset previously discussed suggest this may not be the appropriate method.

PLS overcomes this PCA truncation issue by finding component-pairs between the predictor and target datasets that max-
imize covariance between them. This is in contrast to PCA regression where they are independently chosen to maximize their own explained variance. The algorithm works iteratively, by first finding the covariance-maximizing predictor/response component-pair, the variation captured by these is subtracted from the datasets, and the process repeated. The process is ideally terminated after a sufficient number of components is included so as to explain the true variation in the response dataset. Here we determine this component number using k-fold cross validation.

Figure 7 compares cross-track bias derived from the small-area approximation to that predicted by the regression model. It can be seen that the regression model reduces the noise in the original dataset, and removes some spurious features that extend over multiple cross-track positions that are likely due to real $XCH_4$ enhancements. The updated bias estimate is also less noisy, due to the higher precision of the retrieved ISRF squeeze factors relative to $XCH_4$ used in the small-area approximation.

Figure 8 compares the MethaneAIR $XCH_4$ retrievals over the Midland basin from RF05 before and after the stripe correc-
tion is applied. We have applied the bias correction for a given observation by temporally interpolating the PLS-derived bias (Figure 7, right panel) to the time of observation. In principle, the retrieved ISRF at the observation time could also be directly input into regression model (Equation 7). In practice, we found the former method performed better because (1) ISRF varied smoothly in time and (2) the latter method induced additional noise due to the $10\times$ lower precision of the single-pixel ISRF values compared to the 10s averages used in the PLS model. Figure 8 shows that the bias correction is able to remove the

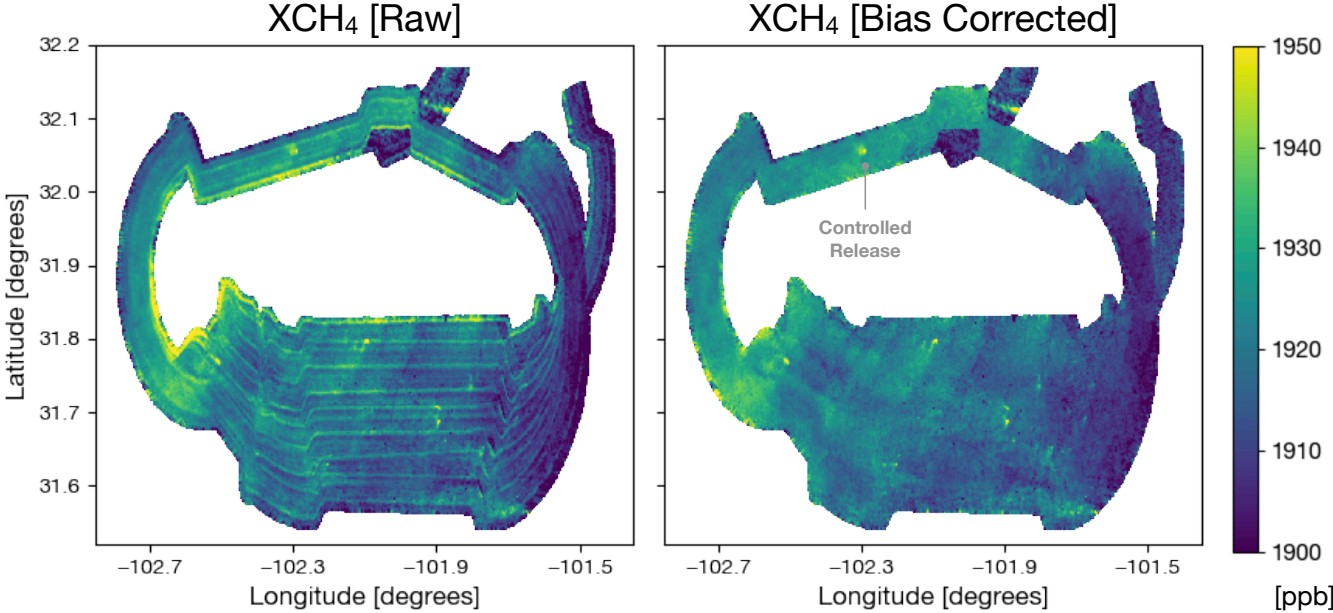

**Figure 8.** Retrieved $XCH_4$ during RF05 (3rd August 2021) gridded at 20x20 m$^2$ resolution prior to (left) and after (right) destriping correction is applied. Here the aircraft is traveling in a clockwise loop, with the northern segments overlapping to target a controlled release.

cross-track striping apparent in the uncorrected data whilst preserving observations of plumes from O&G infrastructure within the basin and at the controlled release site.

## 5   Validation

### 5.1   EM27/SUN Ground Validation

Surveying CH$_4$ across an O&G basin requires 2-3 hours of MethaneAIR observations at its nominal 12km above-ground

observation altitude. It is critical that the retrieved $XCH_4$ is free of significant systematic drifts which could yield artificial gradients within the mapped areas and ultimately reduce the accuracy of emissions inversion. Such drifts are certainly possible given the changes in the ISRF shown in the previous section, and can be quantified using the aircraft overpasses of the EM27/SUN spectrometer sites made over the course of the campaign. Figure 9 shows the comparison between the MethaneAIR and EM27/SUN retrievals for the five flights that intersected the ground sites. For each EM27/SUN overpass

we colocate MethaneAIR retrievals within a 0.05 degree latitude/longitude box of the site location. To remove the influence of the GGG2020 CO$_2$ prior from the comparison, we use the XCO$_2$ observed by the EM27/SUN in place of GGG2020 a priori XCO$_2$ when calculating the MethaneAIR $XCH_4$ from the retrieved vertical column densities ($XCO_{2,0}$, Equation 1). To reduce MethaneAIR and EM27/SUN retrieval differences caused by differences in their respective a priori CH$_4$ profiles and averaging kernels, we adjust MethaneAIR to the EM27/SUN prior and smooth the EM27/SUN observation by the MethaneAIR

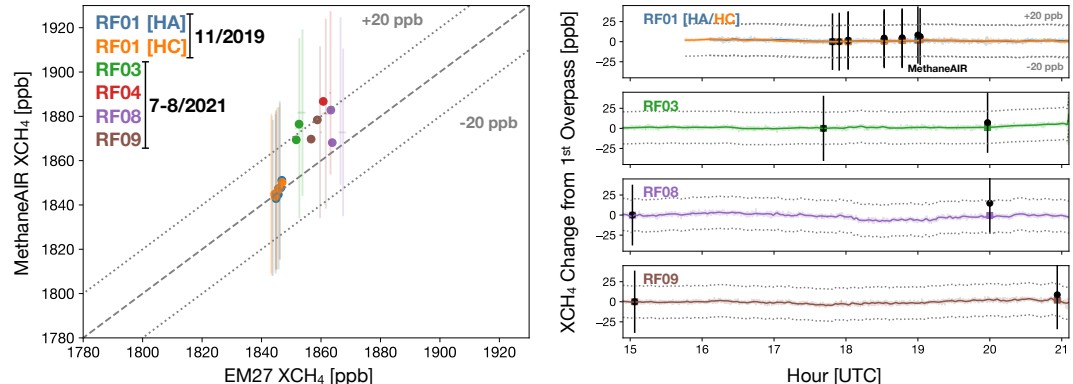

**Figure 9.** Left: Comparison of colocated MethaneAIR and EM27/SUN retrievals of $XCH_4$ (see text for colocation criteria). Error bars represent the $1\sigma$ cross-track XCH$_4$ variability within the 0.05 deg colocation region (since there are $> 3000$ pixels for each MethaneAIR average, the XCH$_4$ random error is negligible). HA and HC refer to the two EM27/SUN spectrometers used in the first flight campaign. $\pm 20$ ppb dashed lines indicate approximate 1% accuracy level. The MethaneAIR XCH$_4$ values have been adjusted to EM27/SUN CH$_4$ profile and the EM27/SUN retrievals smoothed with the MethaneAIR CH$_4$ averaging kernels following Wunch et al. (2011b). Right: The series of the relative $XCH_4$ drift between the MethaneAIR (black dots) and EM27/SUN retrievals (colored lines) after the first overpass for flights with multiple overpasses. The colored squares indicate the EM27/SUN retrievals smoothed with the MethaneAIR CH$_4$ averaging kernels. The $XCH_4$ value at the time of the first overpass is subtracted from each dataset to visualize the relative drift.

averaging kernel, following Wunch et al. (2011b) (Appendix A). Furthermore, we restrict the comparison to retrievals whose *a priori* surface pressure falls within 10 hPa of the pressure measured at the EM27/SUN location, This is more important for the NCAR-Boulder site used for the summer campaign, where there is significant topographic variation associated with the Rocky Mountains immediately west. MethaneAIR retrievals are also screened for poor spectral fits [§], and low signal by rejecting pixels where CO$_2$ and CH$_4$ retrieval degrees of freedom for signal (DoFS) drop below 1, indicating a poor column constraint.

Cloud-contaminated pixels are filtered using the algorithm in Section 3.4.

    In general there is good absolute agreement between the MethaneAIR and EM27/SUN retrievals. The mean bias from the winter flights was 2 ppb, with little drift in MethaneAIR $XCH_4$ for the 5 overpasses over a 70 min interval (Figure 9 right). The mean bias for the summer campaign increased to 13 ppb. This could be partially due to the different EM27/SUN spectrometers used for the summer and winter campaigns, though instrument-to-instrument variations have been shown to be within 0.3%

($\sim$6 ppb) (Alberti et al., 2022). The summer observations were also influenced by visible haze from intense fires in the western US. However the same haze was not visible from greyscale imagery generated from MethaneAIR data, and there is no evidence of strong correlations between retrieved $XCH_4$ and surface albedo, a typical indicator of aerosol presence (Butz et al., 2010). Thus the size-distribution of the smoke aerosols present was likely small enough to not produce strong scattering in the SWIR.

---

[§]Values where the fit residual RMS is greater than 2% are excluded. This is at least 4 standard deviations from the median fit residual of properly converged results thus keeping the majority of good data, whilst removing excluding data from situations where the retrieval is expected to fail, such as over cloud shadows

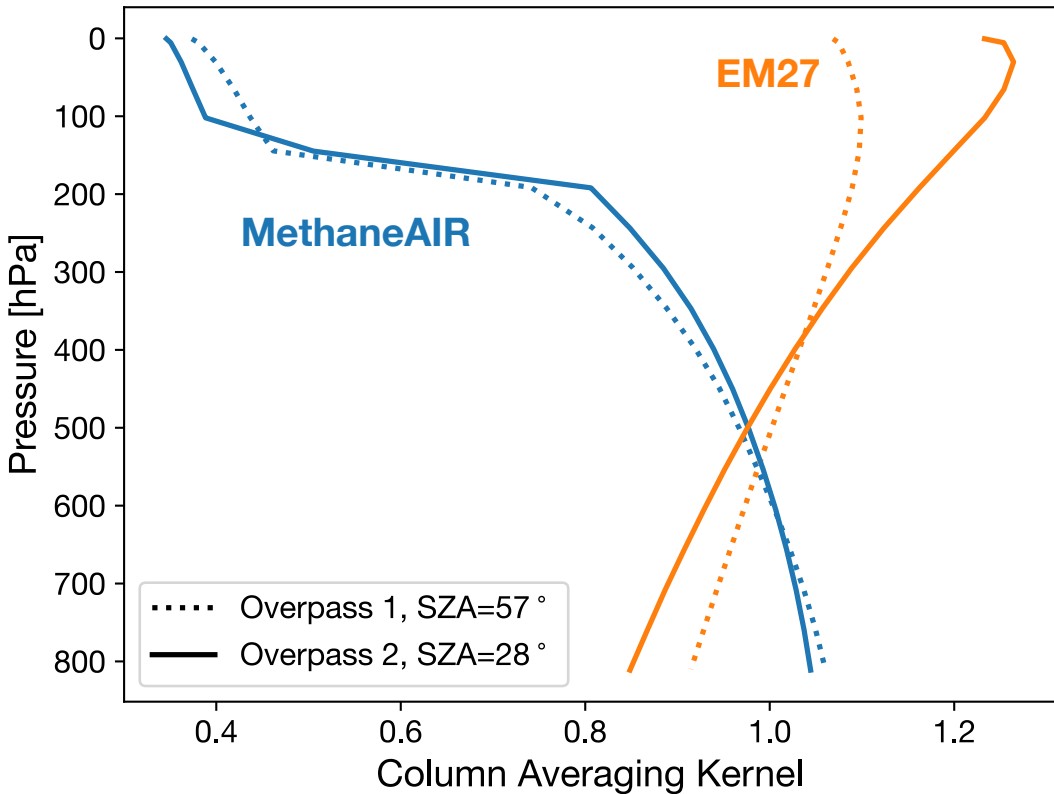

**Figure 10.** MethaneAIR and EM27/SUN column averaging kernels for the two MethaneAIR overpasses from RF08. The MethaneAIR kernels are computed from the average of the retrievals colocated to the EM27/SUN site.

Differences in the vertical sensitivity of the MethaneAIR and EM27/SUN retrievals could explain the flight-to-flight differ-
ences in the summer campaign. Figure 10 shows the column averaging kernels for the two sensors for the RF08 overpasses,
which are representative of observations made at a high $(57°)$ and low $(28°)$ solar zenith angle. The MethaneAIR retrieval is
less sensitive to the airmass above the aircraft, as the solar light returned to the sensor only traverses the layer above once. In
contrast, the EM27/SUN retrieval shows a greater sensitivity to stratospheric differences. The uncertainties in the background
GGG2020 $CH_4$ profiles are largest in the stratosphere, as demonstrated by in-situ, balloon-borne AirCore (Karion et al., 2010)
profile comparisons which produce $\sim 30$ ppb RMSE errors at the tropopause, peaking at $\sim 80$ ppb at $\sim 100$ hPa (Laughner
et al., 2023a). This translates to at least a $\sim 5$ ppb $1\sigma$ variability induced by the different stratospheric instrument sensitivities
in Figure 10, by taking the minimum 30 ppb GGG2020 error, and assuming that it is correlated at all stratospheric altitudes.

The summer flights showed a consistent positive $XCH_4$ drift relative to the EM27/SUN retrievals, ranging from 1.0 ppb/h
for RF09 to 3.2 ppb/h for RF08 (Figure 9, right panel). This drift correlates with the temperature-induced changes in the
ISRF shown in the previous section (Figure 5). The drift is likely induced by the ISRF not being perfectly modeled by the
ISRF squeeze factor. Since the time-evolution of the ISRF squeeze factors showed a similar pattern each flight, this could also

explain why the sign of the drift was always positive. Efforts are currently being made to improve the temperature-stability of the instrument in flight for future campaigns. For now we note that for drifts at these levels it should be possible to infer emissions from diffuse $CH_4$ sources, since the total drift over the time it takes to map a target area (1-2 hours) is approximately an order of magnitude lower than the $XCH_4$ gradients typically observed (see Section 7).

## 5.2 Intercomparison with TROPOMI

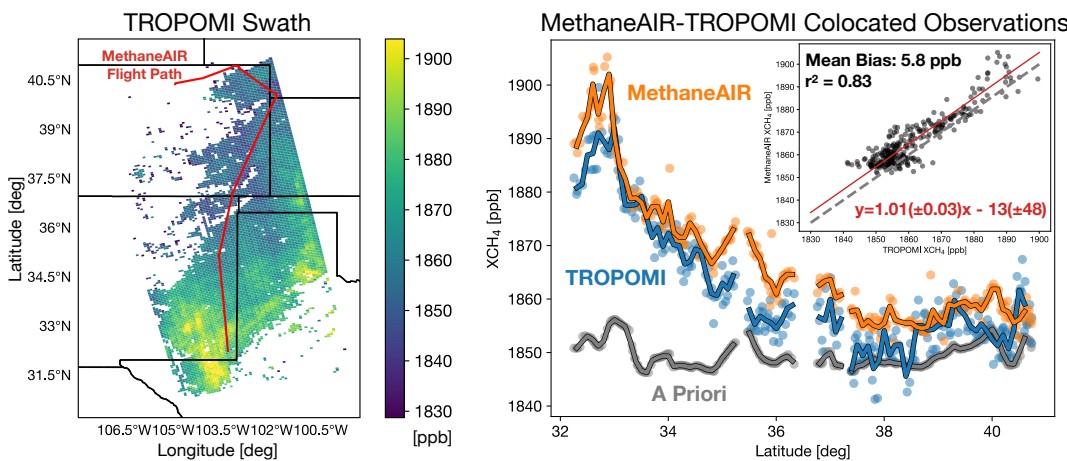

**Figure 11.** Comparison of colocated MethaneAIR and TROPOMI $XCH_4$ retrievals during RF06. Left: TROPOMI $XCH_4$ retrievals (orbit 19769), filtered with quality assurance values > 0.5, destriped using the algorithm described in Liu et al. (2021). The overlapping MethaneAIR flight path within 1-hour of the TROPOMI overpass is shown in red. Right: Comparison with MethaneAIR retrievals averaged to overlapping TROPOMI pixels. The inset figure shows the scatterplot of the matched retrievals, with reduced major axis regression fit (red).

To evaluate a wider subset of MethaneAIR retrievals, we also took advantage of the greater coverage provided by the TROPOMI satellite instrument (Hu et al., 2018). Figure 11 compares TROPOMI $XCH_4$ retrievals from the standard V1 product (Hu et al., 2018) to those from MethaneAIR during RF06, which have been averaged to overlapping TROPOMI pixels within a $\pm 1$ hour time interval. The TROPOMI data have been destriped using the median filter method described by Liu et al. (2021). The overlapping retrievals extend from the Permian O&G basin, and along the transit back to the base airfield in Broomfield CO. MethaneAIR captures the Permian hotspot and the large-scale latitudinal gradient observed by TROPOMI.

The correlation between retrievals is high (RMA regression $r^2 = 0.83$) and absolute mean bias small (5.8 ppb mean bias). The offset between retrievals can be accounted for by the MethaneAIR $XCH_4$ bias induced by the $CO_2$ prior used in the proxy normalization, with uncertainties of $\sim 1$ % in the troposphere, translating to $XCH_4$ retrieval uncertainty of $\sim 14$ ppb. The slope of the regression from the colocated retrievals is 1.01, which is within the uncertainty of the linear regression. Since the enhancement over the Permian basin drives the regression slope, the near unity slope indicates both would yield similar total-basin emission estimates.

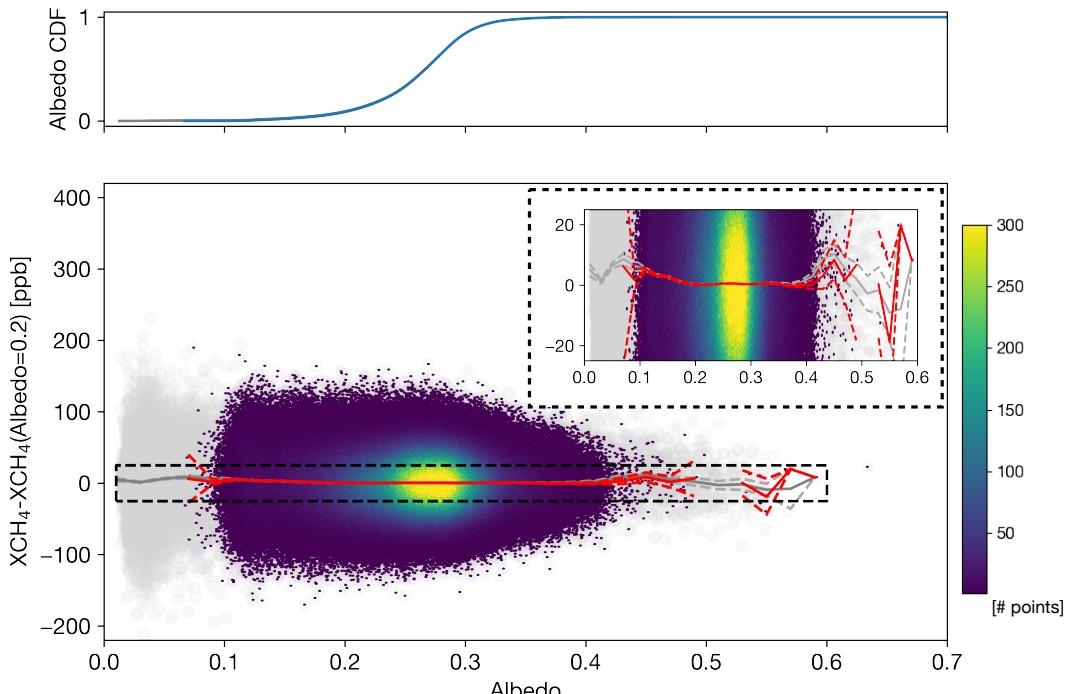

**Figure 12.** Dependence of $XCH_4$ on surface albedo. Relationship between the $XCH_4$ bias as a function of albedo derived using small area approximation (see text) over the return leg of RF06 between 34.8-40°N. The albedo is computed from the radiance at 1622 nm under clear-sky conditions. The $XCH_4$ bias (y-axis) is derived by subtracting the mean $XCH_4$ for albedos between 0.19–0.21. Grey points are those excluded by the cloud-screening algorithm and retrieval quality flags. The grey and red lines show the binned averages computed in 0.02 albedo increments before and after cloud screening respectively, with dashed lines indicating $2\sigma$ bin sample-mean uncertainty. The cumulative distribution function of albedo is shown above the main plot (grey and blue lines all- and cloud-screened-data respectively).

### 5.3 Evaluation against Surface Albedo

GHG retrievals typically correlate with other geophysical parameters, with surface reflectance usually having the strongest correlation (O'Dell et al., 2018; Lorente et al., 2021). The correlation arises due to biases induced by light path modifications from aerosol scattering, which strongly depend on the underlying surface (Butz et al., 2010). For instance, aerosol layers over dark surfaces will tend to shield radiation from penetrating below them, whilst a layer over a bright surface may act to enhance the mean photon path below the layer due to multiple scattering between the surface and aerosol layer. Following Lorente et al.

(2021), we investigate the correlation between $XCH_4$ and albedo by analyzing retrievals over small background regions. We divide the return leg of RF06 between 34.8 and 40 degrees latitude into 3 minute ($\sim$ 30-40 km) segments, small enough to assume $XCH_4$ is constant across the segment (see Figure 11). For the albedo we use the *a priori* reflectance estimate obtained from the observed radiance at 1622 nm. The $XCH_4$ bias in each segment due to surface reflectance is derived by subtracting the

average $XCH_4$ computed from a 0.02 width bin centered at an albedo of 0.2. This value was chosen as it typically corresponds
to where aerosol-induced biases are at a minimum (Aben et al., 2007; Guerlet et al., 2013).

Figure 12 shows the $XCH_4$ albedo-induced biases derived from the return leg. Here the data have been screened using
the cloud algorithm described in Section 5.1. For the screened data there is almost no albedo dependence over the albedo
range 0.2–0.4, corresponding to 90 % of the total observations. The bias then slowly creeps to 5 ppb from 0.2 to 0.1 albedo,
possibly due to residual cloud contamination. This is in contrast to the TROPOMI full-physics retrievals, which show a bias
increase of 1.5 % ($\sim$ 28 ppb) over this range (Lorente et al., 2021, 2023). The stronger dependence in TROPOMI may be
due to the spectral separation of the oxygen A band (757–774 nm) used as the light path constraint from the target $CH_4$ band
(2305–2385 nm), which increases the susceptibility to errors induced from retrieval aerosol optical property assumptions.
Nevertheless, the albedo dependence for MethaneAIR is lower than anticipated, given that during the campaign, observations
were often over regions blanketed by haze from long-range transport of smoke from fires in the Western United States and
Canada. Since a large fraction of this may be in drier air in the free troposphere, the size distribution may be small enough for
the aerosol optical depth to be insignificant at 1600 nm.

Figure 12 also demonstrates the importance of the cloud screening. Below 0.2 albedo the $XCH_4$ bias increases, peaking at
9 ppb at 0.05 albedo. This bias is due to cloud shadows, as well as other low-signal scenes such as those over water, where the
retrieved $XCH_4$ is heavily influenced by the a priori information. The peak is due to the fact there is less spectral information
for constraining $CO_2$, and as a result it tends towards its prior value at a higher albedo than $CH_4$. This can be clearly seen by
the profile DoFS in both species (Figure S3), which drop below 1 at albedos of 0.18 and 0.06 for $CO_2$ and $CH_4$ respectively.
In general the sign and magnitude of this bias will be scene dependent, determined by the *a priori* profile bias along with
additional light-path modifications induced by aerosol scattering. In practice since the degree of regularization is dependent on
radiance, the albedo threshold will also depend on the location and time of measurement. To a first order this will largely be
a function of solar zenith angle (SZA), assuming most surfaces are approximately Lambertian. For the scene shown here the
SZAs ranged from 22-24°, 90 % of the radiance expected for SZA=0°, and the cloud screening and quality filtering remove
points below about 0.1 albedo. Thus we expect the regularization threshold to roughly follow 0.09/cos(SZA), corresponding
to albedo thresholds of 0.13 and 0.18 at 45° and 60° respectively. In practice if these thresholds are too high, the regularization
parameter ($\gamma$, Equation 2) can be re-tuned at the cost of worsening the measurement precision. The choice here has been
optimized to induce no regularization biases over the primary observation targets for this campaign (see Section S1).

# 6 Implications for Plume Identification

## 6.1 Plume Mask Algorithm Description

The previous section showed that the main error in the flight retrievals is random noise. We estimate the precision of the $5 \times 1$
aggregated retrievals of 35 ppb, by taking the standard deviation of the $XCH_4$ retrieved over background locations used in
Figure 12. This value is consistent with our estimate for the native resolution of MethaneSAT (Section S2.1), which has similar
SNR to the $5 \times 1$ aggregated MethaneAIR retrieval. These noise levels reduce MethaneAIR's ability to detect small-scale

$XCH_4$ gradients. To reduce random noise, we apply the Chambolle total variance denoising (TV) filter (Chambolle, 2004) to the retrieved $XCH_4$ images. The TV filter works by minimizing the following cost function between the original ($f$) and smoothed ($g$) images:

$$\hat{g} = \arg\min_{g}\ E(f,g) + \lambda V(g) \tag{8}$$

$$E(f,g) = \sum_{i}\sum_{j}\left(g(i,j) - f(i,j)\right)^2 \tag{9}$$

$$V(g) = \sum_{i}\sum_{j}\sqrt{\left(g(i+1,j) - g(i,j)\right)^2 + \left(g(i,j) - g(i,j+1)\right)^2} \tag{10}$$

$$\tag{11}$$

The cost function penalizes departure from the original image measured by the least squares difference $E(f,g)$. Smoothing is controlled by $V(g)$, which measures the adjacent pixel differences in the image. The degree of smoothing is controlled by the smoothing parameter $\lambda$. The filter was chosen for its favorable properties, such that the resulting smoothed $XCH_4$ fields could be used to estimate $CH_4$ point source emissions ($Q$) without inducing additional bias. Here and for MethaneSAT, the integrated mass enhancement (IME) method (Frankenberg et al., 2016; Varon et al., 2018) is the primary method used to estimate $Q$:

$$Q = \frac{u_{\text{eff}}}{L} IME \tag{12}$$

Here the $IME$ is the integrated $CH_4$ mass of the plume. The effective wind speed $u_{\text{eff}}$ and plume length scale $L$ are parameters that account for impacts of turbulent diffusion on the observed plume extent. $L$ is estimated by taking the square root of the plume area. In practice $u_{\text{eff}}$ is determined from an ensemble of large eddy simulations as a function of the 10m wind horizontal wind speed ($u_{10}$). For MethaneAIR, $u_{10}$ is itself determined by a LES simulation of the target scene at the time of observation (Chulakadaba et al., 2023). The TV filter has some important properties for unbiased estimation of $Q$. First, it is conservative, implying that the smoothing will not bias the $IME$ estimate. Second, it is edge preserving, which means that it will not bias the estimate of $L$.

The IME method also requires an algorithm for plume masking. Figure 13 shows the approach adopted here. First the TV filter is applied to the noisy image. The background $XCH_4$ ($[XCH_4]_{bg}$) is then estimated by taking the $3\sigma$ iteratively clipped mean of the denoised image. An initial mask is computed by flagging mole fractions that are two standard deviations above this background. The final plume mask is computed by discarding flagged pixel clusters that contain less than a threshold number of pixels ($n_{\text{min}}$). This threshold depends on the degree of smoothing by the TV filter.

To optimize the choice of $\lambda$ and pixel threshold we applied the TV filter to a set of 300 plume-free synthetic noisy retrievals, generated from gaussian random noise at the 35 ppb MethaneAIR precision corresponding to the $5 \times 1$ aggregated pixel resolution. Figure 14 (left panel) shows the falsely detected plume mass as a function of the pixel cluster threshold for various levels of smoothing. As smoothing increases, the random artifacts in the image spread in area and decrease in $XCH_4$

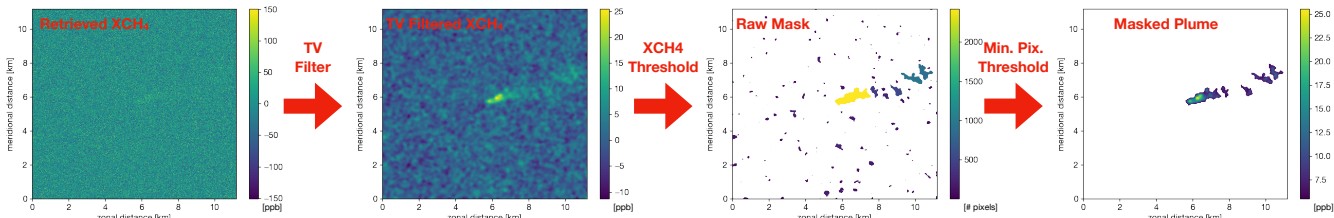

**Figure 13.** Schematic for the plume masking procedure. The left panel shows a synthetic $20 \times 20$ m$^2$ retrieval at MethaneAIR precision of a plume from a 200 kg/h point source simulated with WRF-LES. A Chambolle TV image filter is then applied to the retrieved $XCH_4$ map (middle left). Next a threshold $XCH_4$ is determined from the trimmed mean/standard deviation of the TV filtered image to generate a raw plume mask (middle right). The threshold here is chosen to be 2 standard deviations above the mean. Finally small clusters of pixels (artifacts of the random error in the image) are removed to yield the final masked plume (right).

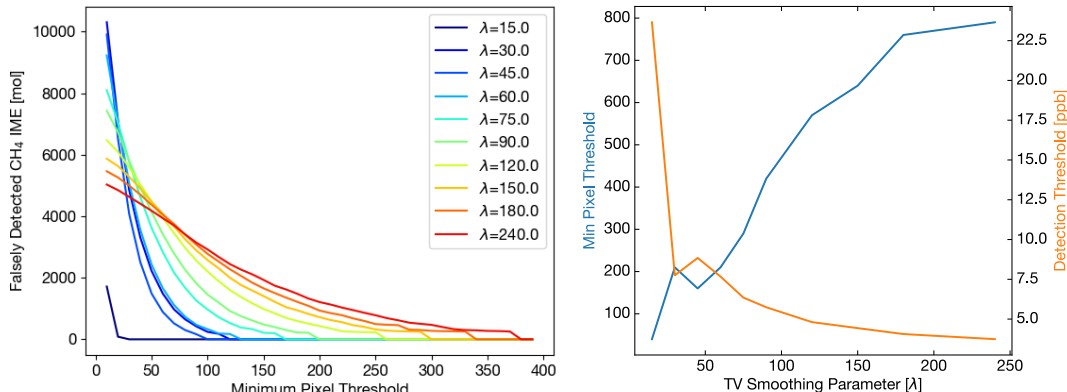

**Figure 14.** Determination of plume mask parameters from plume-free synthetic MethaneAIR retrievals. The left panel shows the falsely detected plume mass for different levels of TV smoothing as a function of the minimum pixel cluster threshold. The right panel shows the minimum pixel threshold required to exclude all falsely detected plume mass as a function of smoothing weight (blue line). The plume $XCH_4$ threshold versus smoothing weight is also shown (orange line).

, lowering the $XCH_4$ detection threshold. As a result, lower levels of smoothing require a higher $n_{min}$ in order to filter the falsely identified mass. The smallest such threshold to fully eliminate the falsely-identified plume mass is shown in Figure 14 (right panel), along with $XCH_4$ detection threshold determined from the variance of the filtered image. At low smoothing levels ($\lambda < 45$) $XCH_4$ detection threshold decreases sharply with only small increases in $n_{min}$. Beyond this the pixel threshold limit continues to increase with little improvement in the $XCH_4$ threshold. We thus set the plume detection parameters at the inflexion point($\lambda = 45$, $n_{min}$=160), which appears to strike a reasonable balance between lowering the $XCH_4$ detection threshold without unnecessary increased smoothing. We repeated the same analysis at the native spatial resolution ($4 \times 20$m$^2$ pixel size,80 ppb precision). The curves look qualitatively similar (Figure S4), with the inflexion point occurring at double the $5 \times 1$ smoothing (plume masking parameters: $\lambda = 90$, $n_{min}$=130).

 **6.2 Application of plume masking algorithm to MethaneAIR**

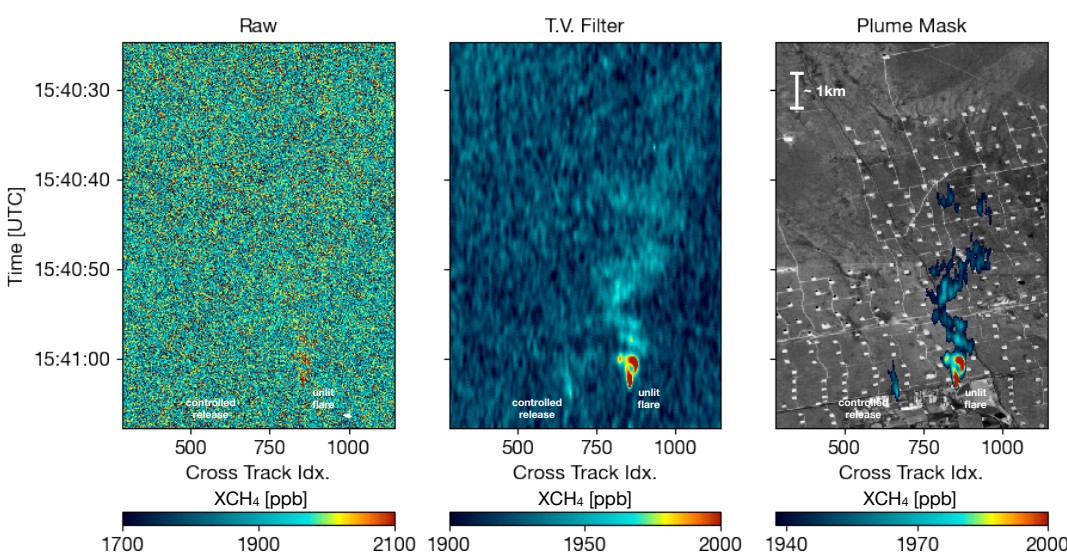

**Figure 15.** Example plume mask applied to MethaneAIR data over controlled release site during RF04. The left and middle panels show the MethaneAIR $XCH_4$ retrievals before and after the Chambolle TV Filter is applied. The right panel shows the masked $XCH_4$ overlaid on a greyscale image derived from the MethaneAIR radiance at 1622 nm. The plume mask detects the O(100 kg/h) plume at the controlled release site as well as a much larger O(1000 kg/h) emission from an unlit flare near the release site.

To see whether the synthetically-tuned filter above works in practice, Figure 15 shows an example of the plume masking procedure applied to real MethaneAIR retrievals at native resolution from RF04. The segment shown occurs over a controlled release site that is being used to validate point source estimation from MethaneAIR. The plume masking procedure successfully masks the emission from the known source, as well as a much larger unexpected source adjacent to the controlled release site 465 from an unlit flare. We estimate a release rate of ~1500 kg/h based on the IME method following Varon et al. (2018), consistent for a source of this type. The full details of the IME approach used here are provided in Chulakadaba et al. (2023).

The masking algorithm detects emissions from the flare up to five kilometers downwind. Whilst this shows that the retrieval performs well, it presents a conundrum: instruments such as AVIRIS-NG (Thorpe et al., 2016) have high $XCH_4$ detection limits but very fine spatial resolution, which enables compact plumes to be detected close to the source. For MethaneAIR the 470 plume $XCH_4$ gradients are smeared out near the source, but detectable over large distances downwind. Thus for retrievals over complex emission fields there is potential for multiple plumes to overlap, complicating point source inversions.

The plume mask $2\sigma$ $XCH_4$ thresholds derived from real data such as in Figure 15 tend to be up to 30 % higher (~ 1.4 ppb) than what we derived from the plume-free synthetic noisy retrievals. Although a portion of this could be explained by as-yet unidentified retrieval biases, meteorological drivers may also play an important role. For instance, the $XCH_4$ retrievals from 475 the EM27/SUN spectrometers during the winter campaign over a relatively clean background region show ~ 1 ppb amplitude

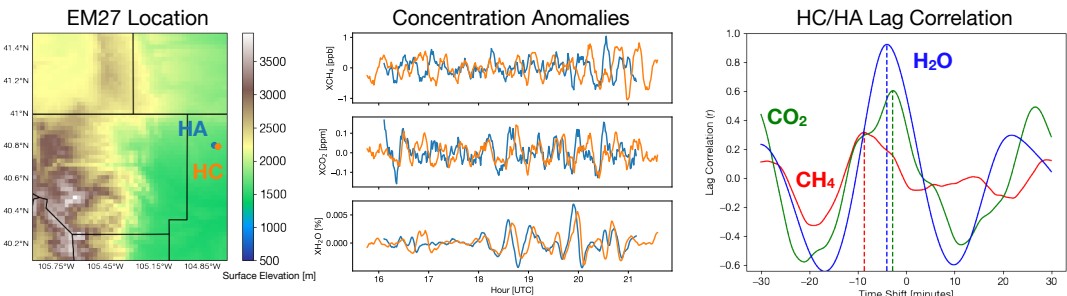

**Figure 16.** Left: Location of the two EM27/SUN Sun spectrometers (HA, HC) used in the winter campaign, overlaid on GMTED2010 digital elevation map (Danielson and Gesch, 2011), Middle: Dry-air mole fraction anomalies of $CH_4$, $CO_2$ and $H_2O$ measured by EM27/SUN instruments HA and HC during RF01 (8 November, 2019). HC was located approximately 1 km downwind of HA (see main text). The anomalies were derived by subtracting data smoothed using a 1-hour moving window from data smoothed with a 5-minute boxcar running window. Right: Correlation between the anomaly time series for different time shifts to the HC time grid, for the segment of HA between 18-20 UTC.

oscillations (Figure 16). In this case the two sites were separated zonally by $\sim$ 1 km, with westerly prevailing surface winds. From the EM27 data, the $XCH_4$ lag-correlation peaks at 6 minutes, and similar values are observed for $H_2O$ and $CO_2$. Such correlations are consistent with the eastward propagation of gravity waves driven by orographic forcing by the mountains to the west of the sites, which could generate the $XCH_4$ anomalies by varying the height of the planetary boundary layer. We

can only expect such meteorological sources of variability to be larger in regions like O&G basins, amplified due to higher boundary layer $CH_4$ mole fractions.

Another source of the higher $XCH_4$ variability in the actual MethaneAIR data could be short-timescale emission variation, which is supported from the observations. Figure 17 shows one such example during RF05, where a plume emanating from a compressor station was observed on the approach to the controlled release site at regular ($\sim$18 minute) intervals. In this instance

a large flash release was observed during the second overpass. Although the flash release had ceased by the third overpass, the remnants of the initial plume are clearly detectable multiple kilometers downwind of the compressor station. Without the context of the earlier observation these pulse releases can be challenging to interpret, and they represent a real source of variation in the data. We identified similar plumes not immediately adjacent to O&G infrastructure during the flights mapping the Permian basin, suggesting source intermittency at time scales less than an hour is common, and highlights that they must

be considered in the emissions inversion of MethaneAIR/SAT data. We see here that they are observable by MethaneAIR and should be detectable by MethaneSAT, based on the size of the observed enhancement and the spatial resolution and precision of both sensors.

### 6.3 Estimation of MethaneAIR's Point Source Detection Limit

Now that we have characterized the instrument performance and defined a plume detection method, the point-source detection

limit for MethaneAIR can be estimated. To do so we use a WRF large-eddy simulation for the meteorological conditions of the

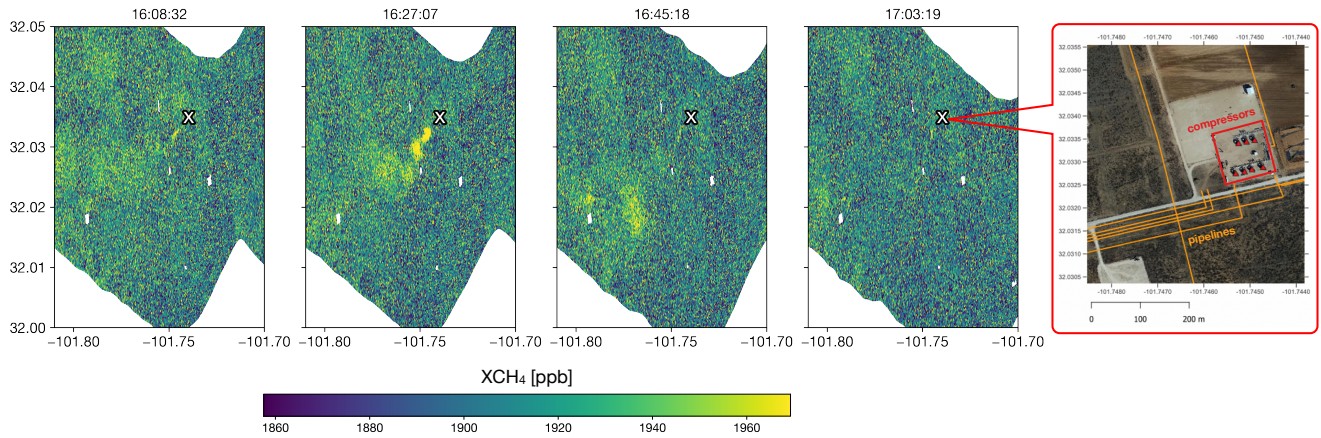

**Figure 17.** $XCH_4$ retrieved over an intermittent source (compressor station) during RF05 (03/08/2021). The title of each panel indicates the observation time in UTC, with the white x marking the source location. The rightmost panel shows a satellite image of the compressor station (©Google Earth), overlaid with locations of pipelines (orange lines) and compressors (red triangles) from Oil and Gas Infrastructure Mapping database (OGIM) (Omara et al., 2023).

controlled release for RF04 (Figure 15), used for the IME calculations. The simulation was performed at 100x100 m$^2$ spatial resolution over a 5.6x5.6 km$^2$ domain, and then interpolated to the 5x1 pixel resolution ($\sim 20 \times 20$ m$^2$). Note that the lower resolution of the simulation will likely lead to an overly pessimistic estimate of the detection limit. We allow the simulation to spin up for 3 h, and then simulate the transport of 5 point sources within the domain for the next 4 h. The mean wind speed

during this period was 2.4 ms$^{-1}$. The simulated $XCH_4$ fields are saved every minute, yielding 1200 plume samples.

A simple estimate of the detection limit can be determined by scaling each plume sample to the minimum criteria where it would be flagged by the plume masking algorithm. From this we determined a median point source emission rate of 93 kg/h (81-109 kg/h interquartile range) for the 5x1 retrieval, using the thresholds corresponding to a TV smoothing parameter $\lambda = 75$. As previously discussed, natural atmospheric variability not due to the plume likely increases the background $XCH_4$ threshold.

If we assume that the 30 % higher threshold found for the real RF04 case is representative, then the median detection limit becomes 121 kg/h (106-141 kg/h interquartile range). This simple bottom up estimate of the detection limit is consistent with the 200 kg/h quantification limit we have determined independently by assessing the performance of the IME method against the controlled release experiments from RF04/RF05 (Chulakadaba et al., 2023). The difference arises in part because here we are referring to detection vs. quantification, as well as differences in the approach used for plume masking in Chulakadaba

et al. (2023).

In order to estimate the accuracy of a point source emission inversion of the MethaneAIR data a full-circle observing system simulation experiment (OSSE) is required, whereby the synthetic MethaneAIR retrievals are created using the WRF-LES and then inverted via the IME method. Varon et al. (2018) performed a comprehensive set of OSSEs for a $50 \times 50$ m$^2$ resolution instrument at various precision levels. At this coarser spatial resolution the precision of MethaneAIR corresponds

approximately to their 1 % precision case, which yielded an emissions uncertainty of 70 kg h$^{-1}$ + 5% of the IME emissions estimate. Note that there is additional uncertainty due to using a reanalysis wind product for $u_{eff}$ estimation, which was estimated to be 15-50 % depending on wind speed. Overall this suggests MethaneAIR is capable of detecting 75 % of US O&G CH$_4$ emissions as point sources, based on estimates from the US Greenhouse Gas Reporting Program (USGGRP) (Jacob et al., 2016; Varon et al., 2018). Whilst there is considerable uncertainty in the USGGRP estimates, the point-source detection rate can be found from the MethaneAIR data alone, as it can also be used to constrain total basin emissions via flux inversions.

## 7   Permian Basin Case Study

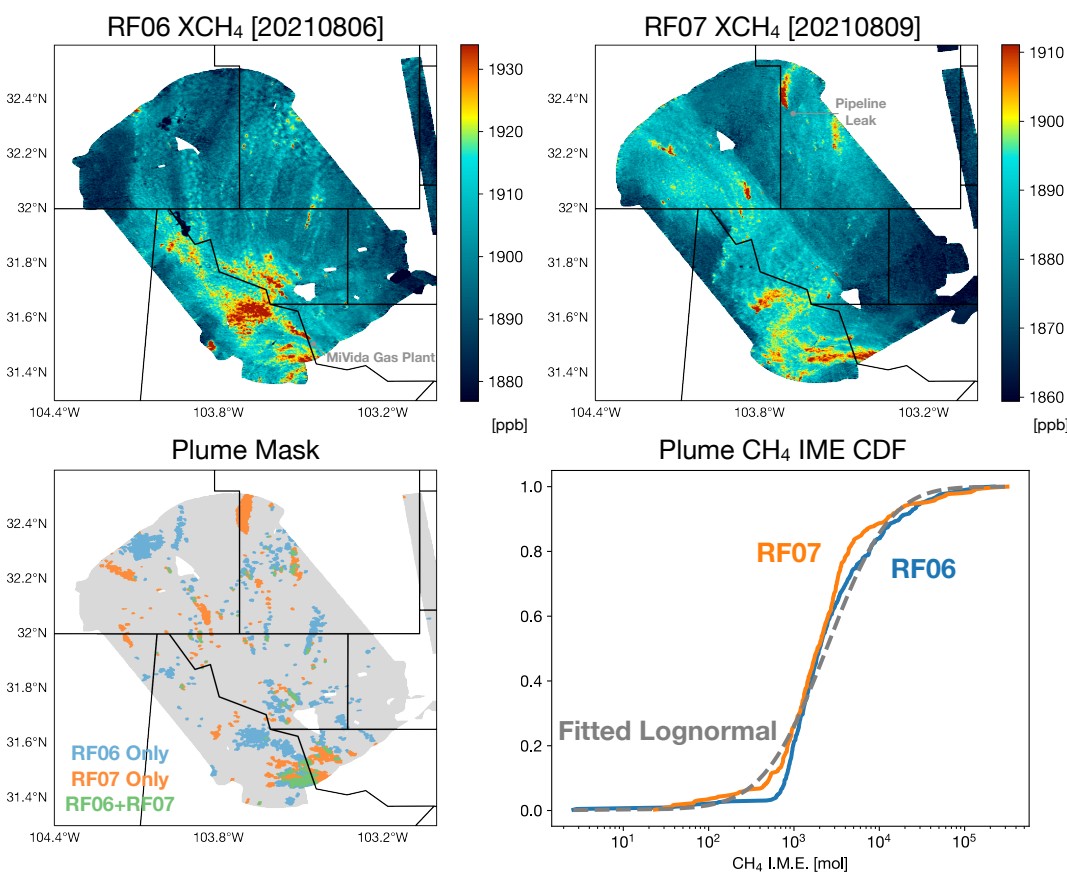

**Figure 18.** Retrievals of $XCH_4$ over the Delaware Basin from two successive research flights. Top panels show gridded $XCH_4$ at 15 m spatial resolution. The gridded plume masks from both flights from the algorithm described in Section 6 is shown in the bottom left panel, with regions detected by one or both flights also indicated. The cumulative distribution of the IMEs derived from the plume masks for both flights are shown in the bottom right panel.

Here we show a case study typical of the flights. Fig 18 shows retrievals made over the O&G infrastructure of the Delaware sub-basin of the larger Permian Basin from RF06 and RF07 (6/9 August, 2021). As the flights were 3 days apart, this provides an opportunity to observe the spatial and temporal variability of sources over a major production region. The flight pattern was the same for both days, with the plane approaching from the north of the basin, before repeating a clockwise oval pattern over the basin that gradually translated northeast each loop (see Figure 1 bottom right). For both days the basin was mapped over an approximately two hour period between 10:00-12:00 local time. The transport patterns for both days are evident from the plumes in the retrieved $XCH_4$ maps, generally south easterly and south westerly for RF06 and RF07 respectively. During RF06, the emissions were clustered around the south of the basin, whereas RF07 shows large sources to the north not present in RF06. In RF07 there is an increase in the $XCH_4$ background over the course of the monitoring period, evident from the discontinuity in $XCH_4$ along the line oriented northwest to southeast through the center of the image.

The sources identified on both flights are shown in the plume mask, verifying that the detected emitters in the basin are highly variable in space and time. The cumulative distribution function for the IMEs from both the flights is also shown in Fig 18, and are heavily tailed as seen in many previous studies (Zavala-Araiza et al., 2015; Brandt et al., 2016; Frankenberg et al., 2016; Cusworth et al., 2021). Despite the spatiotemporal sources being different in RF06 and RF07, their basin-wide point-source probability distributions are similar. If it holds that mean basin-wide emissions are steadier in time, this could allow a reduction in frequency of target monitoring by MethaneSAT, which in turn would allow an expansion of the operational list of targets flagged for regular monitoring.

We now take a closer look at two of the largest emitters identified from the Delaware survey. First we examine the MiVida gas processing plant, where we observed persistent large $CH_4$ plumes for both flights. The plant is located in the southern end of the basin as indicated by Figure 18 (Top left). A total of 9 overpasses were made allowing good characterization of the source (Figure 19) using the IME method. The estimated emissions were persistent and large, ranging from 1858–2518 kg/h upon excluding RF06 16:15 and RF07 17:11 due to poor observational coverage of the target site. Approximately 190 Million cubic feet per day (MMcfd) of gas is processed by the plant according to the US DHS Homeland Infrastructure Foundation-Level Database. With a reported $CH_4$ mole fraction of 0.77 (U.S. EPA Greenhouse Gas Reporting Program (GHGRP)), this implies a leak rate of 1.6–2.1 %. This rate is higher than any of those observed from a comprehensive survey of gas plants (Marchese et al., 2015), and thus suggests a large source of unintended emissions at the plant. From the observed plumes we identified 3 source locations, indicated on the left panel of Figure 19. The largest contributor, indicated by the pink cross, is close to a flare stack and compressor engine shed. The $CH_4$ reported to the EPA GHG reporting program sent to the flares is about half the observed emission (1121 kg/h), indicating that there are other large sources within the plant. There is also a contribution from a set of condensate storage tanks, absorption columns and liquid-gas separators (green cross, Figure 19 left), clearly visible from the corresponding $XCH_4$ enhancement seen in the first pass of RF06. This is consistent with previous surveys of O&G gathering and processing facilities, which observed that venting from tanks tends to be an important source for high emitting sites (Mitchell et al., 2015; Lyon et al., 2016). Finally there is an intermittent source to the south of the plant nearby another set of compressor engines, most clearly visible from the last 3 RF07 overpasses.

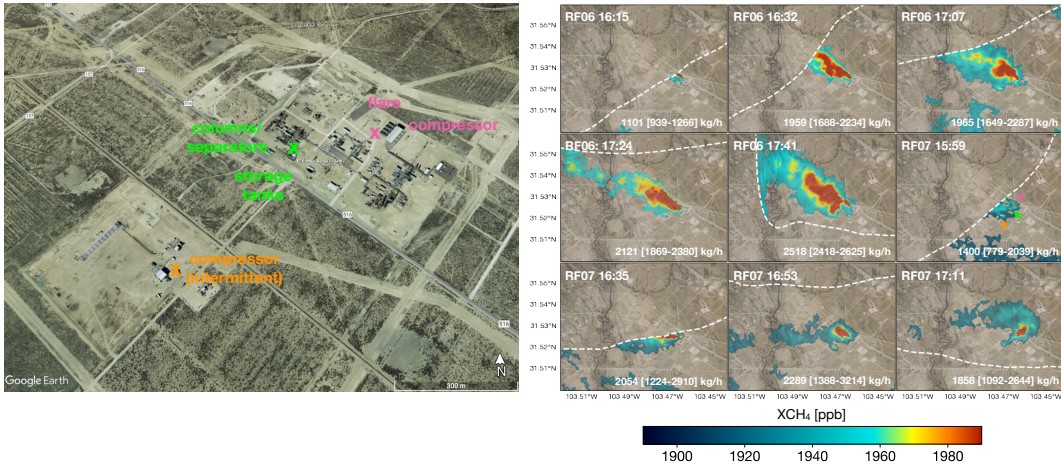

**Figure 19.** $XCH_4$ retrievals from repeated overpasses of the MiVida gas processing plant during RF06/RF07. The left figure shows the ©Google Earth image of the plant, with the approximate location of the three sources contributing to the observed plume within the plant based on the $XCH_4$ retrievals also indicated. The main infrastructure with potentially significant emissions is also labeled. The observed $XCH_4$ plumes from each overpass are shown in the right figure, with the color-coded infrastructure locations shown by arrows in the middle right panel. Dashed lines indicate the edge of the aircraft swath. The emission rates computed using the IME method (Chulakadaba et al., 2023) are also indicated, along with the 2.5-97.5 percentile confidence interval in square brackets. The IME method currently does not account for the impact of partially observed plumes; Such cases will lead to emission underestimates.

The second case described here occurred during RF07, where a large plume extending over 20 km was observed at the northern edge of the Delaware basin (Figure 20 (Top right)). The origin of the plume occurred precisely over an O&G gathering pipeline. The IME-based emission estimate is $\sim 6100$ kg/h (5700-6500)(Chulakadaba et al., 2023), a considerable size for a single source. For comparison (Zhang et al., 2020) estimated Permian emissions of $\sim 2.7$ Tg a$^{-1}$ from TROPOMI, making the pipeline leak 2% the size of average annual emissions from the entire basin. On the 24th of August, 15 days after RF07, an incident report by the pipeline operator was filed (Lucid Energy Delaware, 2021), reporting a rupture along a pipeline weld caused by high line pressure. Based on the operator report on the volume and duration of natural gas vented, we estimated a methane emission rate of 8200 kg h$^{-1}$, which is consistent in magnitude with that observed by MethaneAIR. This case demonstrates that even large-magnitude leaks can go undetected for long periods of time, and suggests that through regular O&G basin monitoring, MethaneSAT will provide an important addition to the existing network of satellites capable of detecting large methane leaks (Irakulis-Loitxate et al., 2022), and help contribute to their mitigation through international activities such as the Methane Alert and Response System (UNEP, 2023).

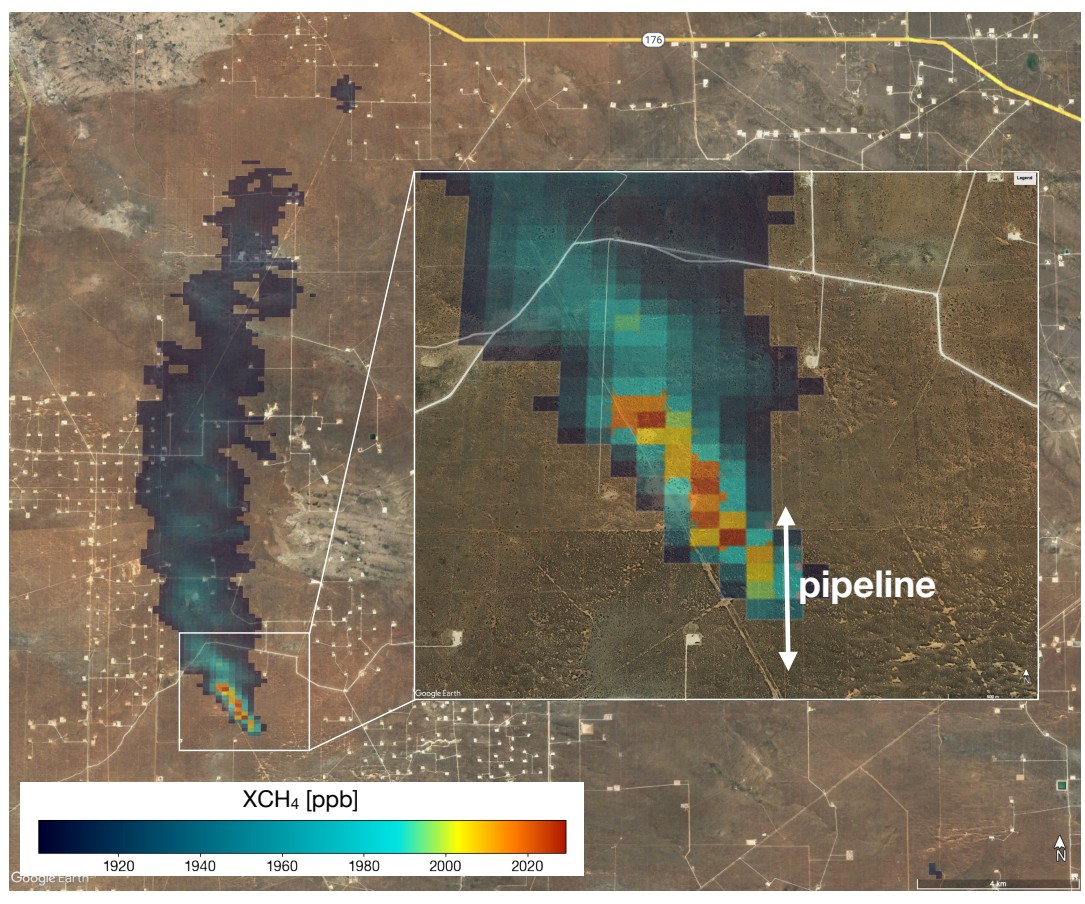

**Figure 20.** Retrieval of a large plume caused by a pipeline leak during RF07, gridded at $100 \times 100$ m$^2$ resolution and overlaid on ©Google Earth imagery.

## 8 Conclusions and implications for MethaneSAT

Here we have described the $XCH_4$ retrieval algorithm designated for MethaneSAT based on the $CO_2$ proxy retrieval approach, and applied it to observations from the first campaign of MethaneAIR, an airborne spectrometer with similar specifications to the future satellite.

Analysis of the flight observations revealed temperature-induced drifts in instrument focus over the course of the flight, typically changing ISRF widths by 10-30 %. We show these can be mostly corrected by fitting a squeeze factor to the tabulated ISRF. We present a PLS-regression based method to remove cross-track biases induced from imperfect modeling of the ISRF defocus using the retrieved squeeze factors. Subsequent validation against ground-based EM27/SUN and TROPOMI retrievals show the instrument is typically accurate to within $\sim 1\ \%$

Based on the flight retrievals we find the instrument precision is typically around 35 ppb ($\sim 1.9$ %) per $20 \times 20$ m$^2$ pixel ($5 \times 1$ pixel aggregate), which we estimate translates to a median point source detection limit of 121 kg/h absent clouds and low albedo pixels, and is in line with the 200 kg/h *quantification* limit determined by Chulakadaba et al. (2023).

Importantly, we find no strong dependence on surface reflectance in the MethaneAIR results. This was a key untested assumption in the early emissions inversions observing system simulation experiments that were used to inform the MethaneSAT instrument requirements (Benmergui, 2019). In those original experiments the assumed precision of MethaneSAT was 0.15% at $1 \times 1$ km$^2$ resolution. In follow up experiments it was found that the satellite could meet its emission constraint goals with that precision at a scale of $5 \times 5$ km$^2$. We expect MethaneSAT to have a similar per-pixel precision as MethaneAIR (35 ppb, Section S2.1). At these levels the 0.15% target precision will be achieved at a scale of $\sim 3 \times 3$ km$^2$[¶], well within the precision requirement. Although the instruments are at different spatial resolution, we do not find a large differences in cloud contamination (Section S2.3.1), or biases caused by sub-pixel inhomogeneities in illumination (Section S2.3.2) and methane concentration (Section S2.3.3) between the two instruments.

The pixel precision/accuracy combination demonstrated here also highlights MethaneAIR's uniqueness in the current set of aircraft sensors, capable of making retrievals at TROPOMI like precision/accuracy at scales of $\sim 100 \times 100$ m$^2$. At these scales new features, such as disconnected CH$_4$ plumes from intermittent sources and $XCH_4$ gradients driven by diffuse sources and boundary layer structures become measurable. As such MethaneAIR provides a valuable testbed to develop new emission inversion approaches accounting for these observable features prior to MethaneSAT's launch.

*Code and data availability.* The L2 retrieval code is available upon request. L2 data is available through the NCAR/UCAR EOL data archive at https://data.eol.ucar.edu/project/MethaneAIR

---

[¶]Calculation based on central limit theorem. 0.15% translates to 2.85 ppb assuming a 1900 ppb $XCH_4$ . The precision of the sample mean reaches the requirement for $\sim 150$ pixels, which translates to $\sim 3 \times 3$ km$^2$, assuming a $140 \times 400$ m$^2$ native pixel size

*Author contributions.* CCM led the study, developed the retrieval algorithm, and wrote the paper with comments/revisions from all authors. SR and JSW contributed to L2 algorithm development and data analysis. SCW, KC, XL, KS, JSB, JEF, YL, MS, and CCM contributed to the mission design (flight planning, instrument requirements). BCD integrated the instrument on the aircraft. JS, JEF, and JH performed the calibration measurements and conducted instrument flights. EC, BL, AHS, KS, and CS developed the L1 calibration algorithms. AC, MS, and JSB performed MethaneAIR $CH_4$ emissions inversions. JLL developed the GGG2020 a priori profile software used by the L2 algorithm. JEF and BB deployed the EM27/SUN instruments. RG and MO contributed facility-level bottom-up $CH_4$ estimates.

*Competing interests.* The authors declare no competing interests

*Acknowledgements.* The Environmental Defense Fund provides primary support for MethaneAIR/SAT project to Harvard, the Harvard-Smithsonian Center for Astrophysics, and University of Buffalo. We gratefully acknowledge assistance from NSF (EAGER grant 1856426 to Harvard University) that supported the flight campaign. JLL was supported by NASA grant NNX17AE15G, with research carried out at the Jet Propulsion Laboratory, California Institute of Technology, under a contract with the National Aeronautics and Space Administration (80NM0018D0004). BCB was supported in part by NASA grant 80NSSC18K0898, NOAA Cooperative Agreement NA22OAR4320151, and NASA/JPL subaward 1615988. We would like to acknowledge technical support of EM27/SUN observations at NOAA/ESRL from Philip Handley, Timothy Newberger , and Anna McAuliffe, as well as Frank Hase for loaning the spectrometer. Finally we extend our appreciation to the MethaneSAT software team (Nick LoFaso, Chris Hairfield, and Sasha Ayvazov) who have adapted the retrieval presented here to a cloud environment for future MethaneAIR campaigns.

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
