# Peer review of "Methane retrieval from MethaneAIR using the CO2 Proxy Approach: A demonstration for the upcoming MethaneSAT mission"

_EGUsphere, 2023_

## Referee Comment (RC2)

**Review of the manuscript**
**"Methane retrieval from MethaneAIR using the CO2 Proxy Approach: A demonstration for the upcoming MethaneSAT mission" by Chan Miller et al.**

**Comprehension**

The manuscript presents shortwave infrared methane measurements using MethaneAIR, an airborne demonstrator for the MethaneSAT satellite. The study focuses on retrieval performance and validation of the CO2 proxy method for CH4 retrieval, addressing instrumental challenges like ISRF drifts, and implementing bias correction techniques. It also validates MethaneAIR's measurements against ground-based data. The preprint aims to extrapolate its findings to anticipate the performance of the forthcoming MethaneSAT mission, particularly in terms of precision and detection limits for methane emissions.

The study presents valuable and relevant research but necessitates revisions, particularly in the description of methodologies and structural organization, to ensure clarity and completeness before it is suitable for publication.

**General comments**

The manuscript's extrapolation of findings from MethaneAIR to MethaneSAT requires improvement, as it frequently lacks comprehensible detail on how this projection is methodologically executed (including the assumptions and limitations).

What types of errors can lower spatial resolution introduce, such as the omission of low-contrast pixels, among others?

The discussion about e.g. the 0.15% target precision for MethaneSAT and its relation to MethaneAIR's performance is limited and I can't clearly see how this extrapolation is methodologically done.

A numerical analysis of the uncertainties associated with emission estimates would be desirable.

Section 8 of the manuscript is relatively brief considering its title, suggesting a need for more extensive coverage and depth in its content.

Certain sections of the paper could benefit from a more concise description, and the overall structure could be improved for clarity and better organization. Consider introducing additional subsections to achieve this.

Consider to display the mathematical expression used to compute the squeeze factor in the forward model. Overall, provide some more details on the forward model (e.g. input quantities, etc.).

Emphasize that the ISRF squeeze is a nominal fit parameter and clarify that the remaining bias is a result of defocusing, which cannot be resolved solely through the ISRF squeeze adjustment but necessitates an additional preprocessing step, such as PLS regression.

The bias analysis would benefit from additional details regarding the underlying assumptions and background information.

**Specific comments**

63: Consider to add some recent literature sources on airborne methane retrieval (e.g. methane by HySpex)

99: Could you provide some brief insights into the command chain involved in operations like near real-time satellite commanding, which relies on forecasts and other factors? I

132: The accuracy of the proxy method could potentially be compromised if there is gas flaring occurring in close proximity?

137: The MethaneAIR ground pixel size significantly differs from the nominal resolution of MethaneSAT. Any conclusions for MethaneSAT need to address assumptions and limitations in such a projection.

185: Consider to be more explicit. But if I understand correctly, there are a total of 19 layers from the Bottom of the Atmosphere (BOA) to the Top of the Atmosphere (TOA), with 13 of these layers situated within the troposphere.

18 + Table 1: Why scale all 19 layers when the enhancement primarily originates at or near the surface and likely remains concentrated in the lower layers for several kilometers, especially under steady wind and stable atmospheric conditions?

199: An Instrument Spectral Response Function (ISRF) squeeze factor of less than 1 typically indicates squeezing, while a factor greater than 1 signifies stretching. So, ISRF squeeze < 1 means squeezing, and ISRF squeeze > 1 means stretching? Be more explicit.

203: Where is this shown?

Eq.(3): Consider to include the complete formula for the squeeze factor.

220: "… high optical depth,…"?

Fig. 4: Is it plausible or reasonable to assume that XCH4 (XCH4 typically refers to column-averaged dry-air mole fraction of methane) is enhanced at the position of the cloud?

264: Is "mechanistically" the right term?

268: "… 10 s of observations"?

Fig. 5: In (a1) and (a2), where is the representation of the XCH4 bias located or depicted?

276: Repeating "valuable"

Fig. 7: Check first sentence in caption. Add XCH4 label to colorbar. Show Beta from Eq. (5) or at least add some information in the caption that helps to better relate the figure with Eq. (5).

318: Clarify.

341: Was the TROPOMI L2 product destriped as part of the processing, or was it delivered already destriped? My assumption is that they provide a destriped product.

369-372: The XCH4 albedo dependence analysis in Fig. 12 is conducted only for a single scene. Is this sentence required?

377: Where to find Fig. S3?

401: Review the sentence (IME is not directly a plume detection method).

Fig. 8: Add more details to the caption. The stripes are in the along-track direction due to the cross-track bias? Do I see multiple parallel flight tracks here?

Fig 9 & Fig. 10: Was the averaging kernel actually considered in the comparison? It may be beneficial to show the formula to accommodate the varying vertical.

Fig 12: Consider providing a brief description of the y-axis label in the caption. Additionally, could you explain why the lines do not exhibit a consistent trend towards either higher or lower values? If scattering is the underlying process responsible for this bias, it's puzzling why it does not consistently manifest as either a positive bias (indicative of multiple scattering) or a negative bias (indicative of single scattering).

390: Please provide details on how the value of 35 ppb was determined. Is it based on the standard deviation or another statistical measure?

Fig 14: Check axis labels on right panel.

545: The resolution is similar to TROPOMI's resolution (at least it is in the same order of magnitude.)

545: Provide more context or specify where the analysis yielding the result of 0.15% is located? Additionally, it would be helpful to understand the assumptions made to arrive at this conclusion.

---

## Author Comment (AC1)

**Authors' Response to Reviews of**

**Methane retrieval from MethaneAIR using the CO2 Proxy Approach: A demonstration for the upcoming MethaneSAT mission**

,
* * *
**RC:** *Reviewers' Comment*,     AR: Authors' Response,     ☐ Manuscript Text

**1. General Comment**

**RC:** *This paper describes the XCH4 retrieval algorithm used for the MethaneAIR airborne remote sensing instrument to detect and quantify CH4 emissions. The algorithm uses the CO2 proxy method to eliminate light path errors due to aerosols. The quality of the retrieved XCH4 maps is assessed, and corrections for changes in the ISRF, a destriping algorithm, and a well-described denoising algorithm are introduced. Additionally, the detection limit of the combination of MethaneAIR and the CO2 proxy approach is estimated. Finally, the emissions of several plumes detected during measurement flights are calculated with the IME method adapted to MethaneAIR. The paper provides valuable information about the retrieval algorithm and post-processing of the retrieved XCH4 maps, with promising results for future flights of the instrument. I recommend publication after the following partly major comments have been addressed.*

AR: We thank the reviewer for their constructive comments and suggestions. Our responses are given below.

**2. Conceptual/Broader concerns**

**RC:** *The background of airborne remote sensing measurements of greenhouse gases needs to be referenced appropriately, with the MAMAP instrument being the first airborne remote sensing instrument to determine emissions from point and small areal sources (e.g., Krings et al., 2011, Gerilowski et al., 2011, Krautwurst et al., 2017, Krings et al., 2018). Relevant work of the airborne remote sensing instruments AVIRIS-NG and GHGSat-Airborne should be included, too. The relevant publications for the satellite remote sensing instruments GHGSat, Prisma, ENMAP, Sentinel-2, CO2M, and Carbon Mapper might also be referenced explicitly, as often a link is made to the satellite system MethaneSat. A corresponding statement on how MethaneSAT fits in this fleet of satellites needs to be included.*

AR: We have updated the text to reference MAMAP

> Here we present results from the maiden flight campaign of MethaneAIR using the operational MethaneSAT $CO_2$-proxy retrieval. This is expected to be the primary retrieval used for emissions inversions. The $CO_2$-Proxy method was first used from an airborne platform by the MAMAP instrument (Krings et al., 2011; Gerilowski et al., 2011), which provided the first remote-sensed estimates of $CH_4$ and $CO_2$ point/small-area sources (Krautwurst et al., 2017; Krings et al., 2018). MethaneAIR builds on this heritage by substantially increasing the sensors rate spatial coverage, mapping approximately $490\times$

the area per unit time relative to MAMAP: This is achieved by its higher nominal operating altitude (12 km vs 1.25 km), faster aircraft speed/exposure time (720 vs. 200 km h$^{-}$1/ 0.1s vs. 0.6s), and its configuration as a push-broom scanner (983 vs. 1 across track pixels). This allows complete mapping of a typical sized oil and gas basin in a few hours of flight time.

The references for most of the aircraft/satellite instruments mentioned were cited but not always explicitly named. We have updated the introduction now to make this clear.

Recently instruments designed to detect high concentrations of $CH_4$ in individual methane plumes have been deployed on aircraft (AVIRIS-Thorpe et al. (2012), AVIRIS-NG-Thorpe et al. (2016), HySpex-Hochstaffl et al. (2023)) and satellites (Sentinel 2-Varon et al. (2021), GHGSat-Jervis et al. (2021), CarbonMapper-Shivers et al. (2021), PRISMA-Guanter et al. (2021), EnMAP-Roger et al. (2024)) to estimate emission rates from point sources.

We believe the introduction already mentions how MethaneSAT fits into the existing set of satellite sensors. We repeat the relevant paragraph here, and attach the reference to CO2M

MethaneSAT has been designed to fill the gap between space-borne global flux mappers and point-source imaging instruments. It contains a pair of 2-D grating spectrometers with a similar spectral resolution to TROPOMI, covering the 1.27 $\mu$m $O_2$ singlet delta ($a^1\Delta_g \leftarrow X^3\Sigma_g^-$) and 1.65 $\mu$m ($2\nu_3$) $CH_4$ bands. Rather than acquiring a broad swath to achieve daily global coverage like TROPOMI and upcoming missions such as CO2M (Sierk et al., 2019), it is maneuverable and built to target scenes at the approximate scale of an O&G production basin ($200 \times 200$ km$^2$ at nadir, for a 30 s collect). By concentrating pixels onto the smaller acquisition area it achieves a pixel size of $130 \times 400$ m$^2$ in its nominal operational mode, enabling $CH_4$ to be retrieved at the accuracy of a TROPOMI like instrument, but with a spatial resolution capable of imaging individual plumes from point sources.

**RC:** *The algorithm is not sufficiently described. It is unclear what mathematical basis is used, especially which regularization and optimizer were chosen (e.g., Gauss-Newton, Levenber-Marquardt, . . . ). Similarly, the convergence criterion is unclear. As this is the first publication of the algorithm, more information should also be added for the forward model, which, to my understanding, is an absorption-only case that should easily be written down as a formula. This is especially important as you do not only use a polynomial for albedo but also for an additive offset if I interpret it correctly (it is never stated explicitly). Why is this additional polynomial included? Do you include a spectral shift fitting parameter in your retrieval? In the current state, the state vector cannot be connected to the forward model in the current state of the description. Finally, I wonder why no theoretical sensitivity study of the algorithm w.r.t. differences in other atmospheric parameters that are not part of the state vector (different albedo types, aerosols, surface elevation, and aircraft altitude changes, as an example) has been conducted. Is a theoretical sensitivity study part of another manuscript currently in preparation?*

AR: We omitted such details as they are generally unusual in the main body of a research article (it is usually left to ATBDs). Since we currently do not have one to refer to, I have expanded Section 1 in the supplement to include such details, including how all state vector variables are included in the spectral fit and optimizer choice.

With regard to the additive offset, it is included to account for broad band stray light, which was present in-part due to the impact of temperature-induced defocusing. We know that it exists based on monitoring signals in the unilluminated part of the detector.

With regard to the retrieval sensitivities, we expect these to be similar to those described in satellite proxy sensitivity studies e.g. Butz et al. (2010) (the geometries and sources of bias e.g. aerosol/surface interactions are not dissimilar due to the high nominal plane observation altitude). In this case the largest difference is probably reduced sensitivity to stratospheric $CO_2$/$CH_4$ concentrations. A sensitivity analysis is part of another manuscript in preparation, the focus of which is comparing proxy normalization by $CO_2$ and from the $O_2$-singlet delta band. We believed it was important to publish this as a reference to the main algorithm, given the public availability of data from the first campaign and the impending launch of the satellite.

**RC:** *For the estimation of the detection limit, there are still a few questions open:*

1. *How do you treat the background? Do you overlay the simulated plume over real data? Or do you also simulate the background?*

2. *How do you treat measurement noise in the simulation? Is it added afterward?*

3. *What is the emission rate assumed for the LES simulation?*

AR:

1. We assume that the background concentration can be subtracted based on the estimate from sigma-clipped mean described in the text. In general there are enough points within the scene such that the random error is negligible. It would only be an issue if a significant portion of the scene was above background levels.

2. Measurement noise (normally distributed, 35ppbv standard deviation) is added on top of the simulation, based on the validation section, which indicated that errors in the retrieval are mostly random, with no strong dependence on surface reflectance

3. The emission rate in the base simulation was $998 \text{ kg h}^{-1}$. However this is not really relevant as $CH_4$ is transported as a passive tracer, so scaling the concentration field is equivalent to scaling the emission and rerunning the exact same simulation.

**RC:** *A thorough uncertainty propagation has to be included for the emission rates and detection limit considerations. The emission estimates in the later parts of the paper are given without any uncertainty range. From the numbers stated in the article, and given that you take the wind from models, there is likely significant uncertainty due to that alone, but also from retrieval precision, and maybe uncertainty in missed parts of the plume may contribute. Please add an uncertainty breakdown to the emission estimates. Also, for IME, you state that you are close to the 1%-case stated in Varon et al., 2018. However, with 1.9% precision, the uncertainty likely is larger than the 70 kg h-1 + 5% stated in the paper. Is an according analysis for a $\sim$2% precision instrument planned?*

AR: We have updated the numbers in the manuscript to include the uncertainty ranges for the emissions estimates for the Mi Vida gas plant and pipeline leak. The emissions estimates have also been updated to reflect the current IME estimates (the previous manuscript used an earlier version). The uncertainties are based upon the the 2.5-97.5 percentile interval of the monte carlo simulation of the errors as described in Chulakadaba et al. (2023). This includes contributions caused by variance in the winds and retrieval errors, but they are difficult to separate due to the calculation method.

The Varon case is 1% precision over a square 50x50m$^2$ pixel. A 1.9% precision at 25x25m$^2$ nominal MethaneAIR pixel resolution does approximately correspond to 1% at the same spatial scale. To emphasize the spatial resolution difference the relevant sentence has been updated for clarity:

> At this coarser spatial resolution the precision of MethaneAIR corresponds approximately to their 1 % precision case,

**RC:** *The paper itself is, in parts, slightly unstructured and jumps between topics. I point out some parts in the line-by-line comments but suggest introducing a few more subsections where the subject of the section covers multiple aspects.*

We have implemented the majority of suggestions in the line-by-line comments. The paper is long, which can make it feel like it does jump between topics. We have added a paragraph to the introduction to better guide the reader through the paper.

> The road map for the rest of the paper is as follows; Section 2 describes the flight campaign, and Section 3 the retrieval methodology. Section 3 describes the method used for retrieval bias correction, needed due to drifts caused by impact of cabin temperature changes during flight. The MethaneAIR observations are validated against ground-based (EM27/Sun) and satellite (TROPOMI) retrievals in Section 4. The results of this validation are used to estimate the MethaneAIR detection limit for point sources, and further discuss challenges for the CH$_4$ emissions inversion problem at fine spatial scales based on the observations (Section 5). We highlight some case-studies from observations in the Permian in Section 6. Lastly, in Section 7 we discuss the implications for MethaneSAT.

**3. Specific Comments**

**RC:** *L32-44: This paragraph might fit better in the later parts of the introduction.*

**AR:** We respectfully disagree. We wanted to state the context early on that is motivating the overall project.

**RC:** *L56-57: I did not find anything in Lorente et al. about the 3% success rate, as well as the causes being the robustness of the full-physics approach. While satellite retrievals usually filter out more than 90% of the data, 3% seems low to me without a source. Additionally, how do the scientific algorithms perform?*

**AR:** I cited the retrieval paper rather than the review paper where this was mentioned (Section 2.2, Jacob et al. 2022) - I have fixed the citation to that.

> Jacob, D. J., Varon, D. J., Cusworth, D. H., Dennison, P. E., Frankenberg, C., Gautam, R., Guanter, L., Kelley, J., McKeever, J., Ott, L. E., Poulter, B., Qu, Z., Thorpe, A. K., Worden, J. R., and Duren, R. M.: Quantifying methane emissions from the global scale down to point sources using satellite observations of atmospheric methane, Atmospheric Chemistry and Physics, 22, 9617–9646, https://doi.org/10.5194/acp-22-9617-2022, publisher: Copernicus GmbH, 2022.

**RC:** *L61/62: The instruments should be named, and MAMAP and GHG-Sat Airborne included and referenced. Additionally, a quick categorization apart from satellite/airborne would be helpful.*

**AR:** The instruments are now named explicitly (see response in Conceptual/Broader concerns). We could not find information about GHGSat-Airborne beyond website press releases. If the reviewer has a reference from the

scientific literature we will gladly cite it. With respect to the "quick categorization," the sentence following L61/62 captures the type of observation

> Such instruments have increased pixel resolution (O(1-10 m) length scale) at the expense of spectral resolution (O(10 nm) vs O(0.1 nm) for global mappers). At these spatial scales $CH_4$ concentrations from plumes originating from point sources can be much higher than the atmospheric background, loosening the retrieval accuracy requirement.

If there is any additional fact pertinent to the discussion can the reviewer please specify.

**RC:** *L66/67: The loosening of the retrieval accuracy is true. However, lower retrieval accuracy comes at the price of higher noise, false positives, and more substantial systematic errors.*

I don't think we are in disagreement - the paragraph states that the retrieval accuracy requirement for such sensors can be loosened for detecting plumes because they can spatially resolve the enhancements. The next paragraph explains the drawback of only targeting plumes, quoting the point source detection limits of such instruments. Implicit in these detection limits are the drawbacks you mention.

**RC:** *L92-94: As the accuracy depends on the SNR (among others) and for smaller ground scenes at higher spectral resolution, the number of photons per detector pixel gets low, a source supporting that statement (e.g., simulation studies for MethaneSAT) would be helpful.*

 AR: We now cite Jacob et al. (2022) which has a comparison in Table 1, The MethaneSAT numbers in that paper are based on a linear sensitivity analysis that was performed when defining the instrument measurement requirements. The ground pixel size was considered in the noise calculation for said analysis.

**RC:** *L114ff: Have the polarization sensitivity measurements been evaluated already in a paper? Or is this planned? This could have a significant impact on the accuracy and precision.*

 AR: We did attempt to do this during RF02 - linear polarizers used also created a lensing effect that caused additional defocusing. It was thus difficult to disentangle the polarization sensitivity from this additional bias. Based on instrument characterization performed from Headwall prior to delivery, the linear polarization sensitivity in the $CH_4$ band was around 5%, with little spectral variation. This would likely only lead to noticeable changes in signal for heavily polarizing reflectance conditions such as water glint, which we currently filter due to detector saturation. The complex polarization wavelength-dependence induced by aerosol/surface reflectance combinations may induce additional bias - when time permits, we do plan to remeasure the polarization sensitivity in lab, and assess potential errors through observing system simulation experiments, to fully assess this.

**RC:** *L131f: The CO2 proxy method for methane for airborne measurements was first applied by Krings et al., 2011, which I suggest adding to the citation.*

 AR: Citation added

**RC:** *L146: The retrieval window for CO2 contains hardly any background information, which would be available in the longer wavelength range for both MethaneAIR and MethaneSAT. Why did you exclude those wavelength ranges?*

 AR: Since the CO2/CH4 windows are jointly optimized, the weaker band is actually included in the retrieval. There is a debate on whether it should be included because there could be a bias between the bands in the spectral database (pers. Comm. w/ MethaneSAT Science Advisory Group). Early in the project we tested fitting the windows independently and did not find a significant difference in the retrieval results, because the

band is too weak to impact the retrieved columns. This is discussed from L175 of the discussion paper.

**RC:** *L148f: Why do you assume that the higher spectral resolution will allow you to only measure from 1597 nm upwards? Did you do any simulations supporting this statement? Otherwise, my previous question is more urgent: Why did you choose not to include the background, especially in the longer wavelength range above 1618 nm?*

AR: The MethaneSAT band starts at 1597 nm (L146), but it has higher spectral resolution. The sentence states that the improved precision from this should partially offset not being able to include the extra 2 nm that are observable by MethaneAIR. I would attach a firm number to this but we are still evaluating the calibration results from TVAC.
Increasing the band width above 1618 does not yield additional information for constraining CO2, but could potentially require additional fit parameters (e.g. extra ISRF squeeze factors), which would degrade the quality of the retrieved CO2 column. The additional state vector elements are well constrained with the current window (all DoFS 1).

**RC:** *L178-L183: While this information is important, it breaks the retrieval description here. I suggest moving the paragraph to a more general "advantages of the retrieval" paragraph at the end of the section. Also, do I assume correctly that the forward model is absorption only? Or does it use aerosol assumptions without optimizing them?*

AR: It is already indicated that the model is absorption only (L176 of discussion). We replace "can be" with "is" to remove this confusion

> Since the proxy method accounts for aerosols via normalization against $N_{CO_2}$ as the light path constraint, $\mathbf{F(x)}$ does not consider scattering and is modeled analytically (Frankenberg et al., 2005).

**RC:** *L186: Does this mean you only optimize the troposphere in the retrieval, i.e., only below the aircraft? Would this change if you fly lower? And do you optimize the 13 levels separately? If yes, how does this work with only 1 – 1.5 DOF for CH4? And how do you optimize/treat water vapor?*

AR: I believe the line already states that the full profile is optimized

> CH4 and CO2 profiles are optimized on a 19 layer vertical grid,consisting of 13 evenly spaced pressure layers from the surface to the tropopause, with a set of fixed pressure levels above.

The 1-1.5 DoFS arise from the a priori constraint to the profile, which specifies a vertical correlation structure between the levels (see Fig. S1). Water vapor is fit as a column scaling factor (Table 1).

**RC:** *L195: "can lead to significant impact on the light focus on the detector" – could you please clarify what you mean by this and why the F-number is important here?*

AR: The F-number is the ratio of the focal length to the aperture diameter of the entrance optic. It is a measure of the light collecting efficiency of the instrument. Small F-number instruments will capture more light, but are susceptible to defocusing. This is the result of the focal length in the entrance optics being relatively short, such that a small change in relative position of the optical components from say thermal expansion, can significantly move the system away from the ideal image point.

**RC:** *Table 1: The Albedo a priori is stated to be the MethaneAIR Radiance – averaged over which wavelength range? Also, an albedo uncertainty of 100% is very small for low albedos, e.g. 0.05, while it is pretty large*

*for typical albedos of 0.2-0.3. Is this on purpose? Finally, why do you include a radiance offset in the state vector, which is an additive offset, I assume?*

AR: A footnote has been added to Table 1 explaining how the radiance is used for the albedo calculation

> Albedo estimated from 5-wavelength-pixel average centered at 1622.5 nm.

The coefficients from the Albedo DoFS are always $\sim 1$ so in practice the variation does not have a large impact on the retrieval results. Low-albedo scenes are also removed by quality filtering.
The radiance offset is additive. It is included to account for broad-band stray light, which also appears to stem from the instrument defocusing issues. As one piece of evidence, we have observed detector signals in the unilluminated guard-pixel region correlating to retrieved squeeze factors.

RC: *Fig 3: Interestingly, the RMS after correction is lower at regions on the chip where the defocus is more prominent (as indicated by the larger Squeeze factor). While it aligns with the L1 prediction (how is it calculated from the radiance uncertainty?), it is still interesting that a lower resolution seems to perform better than a higher resolution (provocatively speaking).*

AR: The L1 prediction can be calculated from radiance uncertainty. The relationship between the observed $y_i$ and modeled ($F_i$) at wavelength pixel $i$ is

$$y_i = F_i + \epsilon_i \tag{1}$$

$\epsilon_i$ is the radiance error, and is assumed normally distributed ($\epsilon_i \sim \mathcal{N}(0, \sigma_i^2)$). The RMS (normalized by mean radiance $\overline{y}$) is given as

$$RMS = \frac{1}{\overline{y}} \sqrt{\frac{1}{n} \sum_{i=1}^{n} (F_i - y_i)^2} \tag{2}$$

$$= \frac{1}{\overline{y}} \sqrt{\frac{1}{n} \sum_{i=1}^{n} (\epsilon_i)^2} \tag{3}$$

The predicted RMS is the expectation of the above

$$E[RMS(\boldsymbol{\epsilon})] \approx \frac{1}{\overline{y}} \sqrt{\frac{1}{n} E\left[\sum_{i=1}^{n} (\epsilon_i)^2\right]} \tag{4}$$

$$= \frac{1}{\overline{y}} \sqrt{\frac{1}{n} \sum_{i=1}^{n} E\left[(\epsilon_i)^2\right]} \tag{5}$$

$$= \frac{1}{\overline{y}} \sqrt{\frac{1}{n} \sum_{i=1}^{n} \sigma_i^2} \tag{6}$$

Equation 5 assumes that the $\epsilon_i$ are independent of one another, and Equation 6 follows from the 2nd moment of the normal distribution. Moving the expectation operator (Equation 4) inside the square root basically assumes that the width of the distribution of the error residual sum ($E\left[\sum_{i=1}^{n} (\epsilon_i)^2\right]$) is fairly narrow to not incur a large error. This is a reasonable to assume because of the large number of variables summed ( 500), so

one can make a argument based on the central limit theorem that the distribution should approach a narrow Gaussian. When making the estimate I quickly tested this by running 100 monte carlo trials and it produces almost identical results.

[Figure]

The RMS is lower in the regions of greater defocus, however the "L1 prediction" derived from the L1 radiance noise estimates also shows a similar trend - so this is likely related to the radiance pixel precision being better on that side of the detector.

**RC:** *L213ff: Why do you use the OCO-2 algorithm a priori uncertainties? Especially for H2O, I assume some difference in uncertainty between a satellite covering the whole earth with the natural variability of water vapor globally and the local variability of water vapor during an airborne remote sensing flight of 2 hours. The same holds for the surface pressure uncertainty. And why use the UoL GOSAT Proxy retrieval covariance matrices for the CH4 and CO2 profiles and not covariances derived from the GGG2020 Priori Profile Software results?*

AR: For H2O there is enough information to retrieve the column at the current a priori uncertainty, so relaxing it further would not appreciably change the results. We employ a high-resolution ( 20m) surface elevation database to adjust the GEOS-5 surface pressures. Based on our experience retrieving surface pressure from the $O_2$ band, it appears that this database is capable of accounting for small-scale variations in pressure. GGG2020 does not contain gas profile uncertainties, it only estimates background CO2/CH4. The actual GGG retrieval optimizes column scaling factors, so there is no information specifying vertical CH4/CO2 correlations required to adequately constrain the profile solution.

**RC:** *L244: "temperature changes can defocus light at the FPA" – what do you mean by that? A defocus of the image in the FPA plane? How does this happen mechanically?*

AR: The refractive indices of elements in the optical chain change with temperature, leading to a defocusing effect.

Thermal expansion can also change the relative positions of these elements, yielding a defocused image.

RC: *Figure 5: Interestingly, the largest biases and bias changes do not appear in the regions with the largest ISRF change. Or is this the bias, including the ISRF fit? Then, my question would be why the bias is largest in the area where the lowest correction was necessary.*

This is more likely to do with how we have defined the cross track bias. The baseline for the bias data in Figure 5 is the mean of all the $XCH_4$ cross-track medians. Thus the data at the higher cross track indices may be closer to the truth, and everything else biased negatively. The general issue is that we cannot know the absolute $XCH_4$ from the small-area approximation alone, so we must make some assumption, taking the mean being one of them. The idea of of choosing the region with the lowest-correction is an interesting one though - we will likely evaluate it in the future.

RC: *Figure 6: First, I am curious about the dips/jumps in temperature. Do they correlate with some in-flight operations?*

AR: The dips correspond to the times where dark current measurements were taken for spectral calibration. This involved the operator applying the telescope lens caps manually, which disturbs the environment around the instrument. After the first campaign a shutter system was added to the system which removes these temperature blips.

RC: *Second, I suggest adding a shifted orange (dashed?) line to visualize the correlation better so that the steepest gradient overlaps with the temperature curve.*

AR: The actual value of the correlation is already shown in the inset, which is more quantitative than visually shifting the orange curve. Since the figure labels are small, I have updated these to show what the inset is more clearly.

RC: *L250: 10 s corresponds to approximately how many spectra? I.e., how robust is this median over, e.g., plumes? Do you do that as a running median, or each 10 s step is condensed to one median value?*

AR: 10s corresponds to 100 spectra, which covers a distance of 2km along-track. The median is performed stepwise i.e. once for each 10s segment. In most cases this is robust over plumes, however in unlucky scenarios such as when the plume is oriented in the flight direction this could cause a bias. However the point of doing the PLS regression, which locks the XCH4 bias correction to variations in the ISRF squeeze factors, is intended to remove such cases.

RC: *L254: "The relative XCH4 cross track bias is then derived by subtracting the mean of all the cross track background values." I do not understand this. Can you provide a formula for the bias in cross track pixel i at time t?*

Equations are added to the text for clarification

Let $XCH_{4,g}(ix, it)$ be the retrieved $XCH_4$ for 10 s segment $g$ at cross/along track index $ix$ and $it$ respectively. The cross-track bias at cross track $ix$ for the granule $(B_g(ix))$ is estimated as follows:

$$XCH_{4,g}^{med}(ix) = \text{med}\left(XCH_{4,g}(ix, :)\right)$$
$$B_g(ix) = XCH_{4,g}^{med}(ix) - \text{mean}\left(XCH_{4,g}^{med}(:)\right)$$

In the above med and mean denote the median and mean respectively, and : denotes the array axis in which the operation is taken over.

**RC:** *L257: "is a common feature of other 2D grating spectrometers." – please cite according examples.*

AR: Added example citations for MODIS, OMI, and TROPOMI

The relatively constant pattern is likely due to instrument slit inhomogeneities, and is a common feature of other 2D grating spectrometers (e.g. MODIS: Rakwatin et al. (2007), OMI: Boersma et al. (2011), and TROPOMI: Borsdorff et al. (2019)).

**RC:** *L259-263: The second ISRF PCA score has a totally different time evolution than the first ISRF PCA score. The same holds for the XCH4 PCA scores. Therefore, if the second ISRF PCA score has a similar correlation with both XCH4 PCA scores, I would be hesitant to call the correlation between the first components high. For completeness, what is the correlation between the first ISRF PCA score and the second XCH4 PCA score?*

AR: The Pearson $r$ between the first ISRF PC score second XCH4 bias PC score is -0.40. The second PC does not have a significantly different time evolution if multiplied by $-1$ (the sign of the score is somewhat arbitrary as you can multiply the score and PC by -1 to yield the same result). In this case the curves look similar but the peak in the second lagging by 2 hours. We are calling the coefficients high purely based on the pearson correlation coefficients calculated from the scores.

**RC:** *Equation 5: Please expand this to include the complete formula from the pixel segments to the pixel bias.*

The current approach simply time interpolates the ISRF predicted bias in Figure 7. It is stated in the text

We have applied the bias correction for a given observation by temporally interpolating the PLS-derived bias (Figure 7, right panel) to the time of observation.

**RC:** *L258ff: It might be worth putting the across-track bias correction part (which you later also call "destriping") into its own subsection and separating it from the general ISRF fit result analysis. Additionally, state clearly at the beginning that the bias correction is to remove across-track biases, i.e., stripes. Do you have to calculate each flight's correction separately, or is the correction found for one RF sufficient?*

We have split the section into two segments - "Evidence of time-dependent cross track bias" and "MethaneAIR cross track bias correction algorithm"

**RC:** *Figure 8: While the correction largely removes the striping, there is still some residual track-to-track bias (and, in some cases, some residual stripes) left over. Did you quantify this?*

The residual drift is later quantified by the EM27/Sun comparison, where the influence of defocusing is noted

The summer flights showed a consistent positive $XCH_4$ drift relative to the EM27/SUN retrievals, ranging from 0.9 ppb/h for RF09 to 3.1 ppb/h for RF08 (Figure 9, right panel). This drift correlates with the temperature-induced changes in the ISRF shown in the previous section (Figure 5). The drift is likely induced by the ISRF not being perfectly modeled by the ISRF squeeze factor. Since the time-evolution of the ISRF squeeze factors showed a similar pattern each flight, this could also explain why the sign of the drift was always positive.

In future flights we may consider allowing $XCH_4$ drift in the PLS regression. This needs careful evaluation as it could subtract real variation if say the background was increasing in time.

**RC:** *L305f: How do you rescale the MethaneAir retrievals with the EM27/SUN XCO2? Are you using the EM27 XCO2 instead of the model XCO2 in the proxy formula?*

 AR: Correct, changed the line to state this explicitly:

> To remove the influence of the GGG2020 $CO_2$ prior from the comparison, we use the $XCO_2$ observed by the EM27/SUN in place of GGG2020 when calculating the MethaneAIR $XCH_4$ from the retrieved vertical column densities ($XCO2_0$, Equation 1).

**RC:** *Figure 9, right side: are the lines and dots shifted such that the value is 0 for both dot and line at the first dot?*

 AR: Yes - updated caption to explain this

> The XCH4 value at the time of the first overpass is subtracted from each dataset to visualize the relative drift.

**RC:** *L309: Please provide threshold values for the filters and how and why you chose these values.*

 AR: Updated section with concrete numbers

> Updated section with concrete numbers "MethaneAIR retrievals are also screened for poor spectral fits [a], and low signal by rejecting pixels where $CO_2$ and $CH_4$ retrieval degrees of freedom for signal (DoFS) drop below 1, indicating a poor column constraint. Cloud-contaminated pixels are filtered using the algorithm in Section 3.4."
>
> ———————
> [a]Values where the fit residual RMS is greater than 2% are excluded. This is at least 4 standard deviations from the median fit residual of properly converged results thus keeping the majority of good data, whilst removing excluding data from situations where the retrieval is expected to fail, such as over cloud shadows

**RC:** *L313f: "This could be partially due to different EM27/SUN spectrometers used...": Should that not at least partly be mitigated by rescaling the retrieval with the EM27 XCO2?*

 AR: In practice, scaling the MethaneAIR XCH4 by EM27 XCO2 scales the aircraft observations by the EM27 XCO2 bias. To mitigate the effect of instrument-to-instrument differences the EM27 XCO2 bias would need to be positively correlated with the EM27 XCH4 bias. In general the scaling differences seem random instrument-to-instrument (See Alberti et al. (2022) EM27/Sun intercomparison, Figure 24).

**RC:** *L339: Why did you choose the standard V1 product, assuming you mean the operational product and not a newer version? As far as I know, significant improvements have been made since this version, and they might affect the comparison. Also, while the uncertainty due to the proxy normalization is sufficient to explain the bias, it might be that in the TROPOMI results, there is still a bias, which could either improve or worsen the comparison.*

 AR: At the time of comparison that was the latest version (we have been slow to publish). The comparison with V1 was already quite good, and was intended as a sanity check for MethaneAIR rather than the other way around.

**RC:** *L358: Why use the a priori estimate and not the fitted albedo?*

AR: The albedo in the plot is the lambertian reflectance derived from the L1 radiance - In the non-scattering forward model the retrieved albedo is almost identical and would not change the conclusions drawn from the analysis.

RC: *P21, second paragraph: Likely, this paragraph makes much more sense once the optimal estimator and regularization are named. However, as of now, it is a bit confusing, as it is not fully clear what the goal of the paragraph is.*

AR: We are only saying that the results below that albedo will be heavily influnced by the prior due reduced signal. We have changed the phrasing to make this clearer.

> This bias is due to cloud shadows, as well as other low-signal scenes such as those over water, where the retrieved $XCH_4$ is heavily influenced by the a priori information.

RC: *Heading to 6.1: In this paragraph, additionally, IME is partly introduced, which should somehow be either reflected in the heading or, preferably, the introduction to IME is moved to a separate subsection.*

AR: We only introduce the IME method here because it is needed to discuss the reasons for choosing the total variance filter as part of the plume masking algorithm. The section only covers plume masking and point source detection limits.

RC: *L400: It sounds like the filter was already implemented by Frankenberg et al., 2016 and Varon et al., 2018, which, to my knowledge, is not the case. Is there a source where the advantages of this filter to IME are studied? If not, please provide some context for why it is advantageous here.*

AR: The additional image-processing filter is required as we do not spatially resolve the plumes as well as the aforementioned sensors (pixel size O(10m) instead of O(1m) - at the finer scale the plumes are more apparent, whereas at the scale of MethaneAIR the plumes themselves are comparable in magnitude with the instrument single-pixel precision. However, since the retrievals do not have systematic biases near the magnitude of the AVIRIS-like sensors, this permits the sort of spatial-averaging that the image-processing filter applies, to reveal coherent plume structures that can then be masked for use with the IME method.

RC: *L406: Ueff is, to my knowledge, determined from the u10 10-meter wind from models. This is done via LES, but not for each target scene, at least in the papers you cited. Also, the MethaneAIR inversion paper (Chulakadabba et al., 2023, disc) states that ueff is a function of log(U10) + 0.6. Please also cite Chulakadabba at the first mention of the IME approach applied here.*

Yes you are correct for the standard IME method - the relationship between $U_{10}$ and $U_{eff}$ is determined from an ensemble of LES simulations not for the specific target. In the approach used for MethaneAIR (Chulakadaba et al., 2023) the U10 used in this calculation is actually that calculated as the RMS of the 10m wind speed from an LES simulation of the target $\pm 15$ minutes either side of the observation time. Based on comparisons with the controlled releases from RF04/RF05, this approach outperformed using the met product used for the LES models boundary conditions, and the actual wind speed measured at the site. We have updated the paragraph to clarify this

> Here the $IME$ is the integrated $CH_4$ mass of the plume. The effective wind speed $u_{\text{eff}}$ and plume length scale $L$ are parameters that account for impacts of turbulent diffusion on the observed plume extent. $L$ is estimated by taking the square root of the plume area. In practice $u_{\text{eff}}$ is determined from an ensemble of large eddy simulations as a function of the 10m wind horizontal wind speed ($u_{10}$). For MethaneAIR, $u_{10}$ is itself determined by a LES simulation of the target of scene at the time of

observation (Chulakadaba et al., 2023).

**RC:** *L411f: What is the "3-sigma iteratively clipped mean"?*

AR: It is a sigma-clipped mean - just a robust estimator for the actual mean, which I thought was widely known e.g. `https://www.gnu.org/software/gnuastro/manual/html_node/Sigma-clipping.html`

**RC:** *P23 last line: Please include the curves for the native resolution, maybe in the supplement.*

AR: Added to supplement

**RC:** *L435-439: While this is a real consideration, the longer plume detection raises additionally the question of how to treat plumes cut off by the swath edge in the IME approach like the plumes in Fig. 19. This gets additionally important for highly variable plumes or puff releases.*

Correct. Currently plumes cut at the edge of the swath are underestimated by the IME method for obvious reasons. We currently just acknowledge these as source detections rather than quantifications. The problem with puff releases is discussed later in the section. The overall intention of the section is to highlight these unintended emission inversion challenges that come out of having a detector with both relatively high spatial resolution and precision.

**RC:** *L458: This is a very strong statement, especially as no MethaneSAT data yet exists. I suggest rephrasing that such detached plumes should be detectable by MethaneSAT.*

AR: Updated:

> We see here that they are observable by MethaneAIR and should be detectable by MethaneSAT.

**RC:** *After Line 460, I suggest the beginning of a separate subsection, as this is a significant point of the paper.*

AR: Added subsection heading: "Estimation of MethaneAIR's Point Source Detection Limit"

**RC:** *Figure 17: Please enlarge the red cross, and maybe color it differently (possibly black), as it is hardly visible and due to the blue in the figure also slightly hurting the eyes. Similarly, the red triangles in the Google Earth overlay are hardly visible. Maybe mark the whole region of compressors with a box?*

Changed x to white with black outline based on this stack overflow post. The red box bounding the compressors has been added to the figure as suggested.

**RC:** *L464: You state that the lower resolution of the simulation will lead to an overly pessimistic estimate of the detection limit. What is the reason for that?*

AR: Effectively the simulation is lower resolution than the data - the plume is artificially smoothed by this. Thus the enhancements above the background that are detectable will also be harder to detect, purely as a result of the simulation being coarser than what can be resolved by the sensor.

**RC:** *L466: Why do you assume such a low wind speed? The detection limit is linked to wind speed, too, and wind speeds of 5 m/s and higher are not unusual, especially in cloud-free conditions over sand or shrubland, not uncommon for O G production sites.*

AR: The WRF simulations were performed for the controlled release study in Chulakadaba et al. (2023). That happened to be the wind speed. To first order, the concentration field (and hence detection limit) will scale linearly with wind speed (under the simple assumption of advection-only transport), implying all else being

equal, a detection limit of 260 kg/h at 5 ms$^{-1}$. However as you mentioned vertical stability and surface roughness are going to complicate this estimate due to their impact on turbulent transport.

RC: *L467: How exactly do you scale the plume sample? Do you multiply the base emission and all XCH4 values by the same constant? Is this realistic?*

AR: This is exactly what would happen in the LES simulation if it were run identically with the emission rate scaled by the same constant. The model is linear this way.

RC: *L483: What is the uncertainty range on the 75% number of detectable sources? As this is quite a high number, and the wind speed uncertainty alone accounts for 15-50%, plus 70 kg h-1 emission uncertainty, which for sources near the detection limit is also nearing 50%, I assume this number should have quite a substantial uncertainty.*

AR: Not the fraction of detectable sources - the fraction of total OG CH4 emissions that are theoretically detectable as point sources, as indicated in the text. Tautologically, larger sources are going to contribute more to the total emissions. This is why the 75% number appears high. It is also subject to the uncertainty in the EPA USGGRP estimates - however it is the best bottom-up source for the entire US that we currently have found. This is also already acknowledged in the paragraph.

RC: *L500: Is there a reason why Frankenberg et al 2016 is not on the list, as you cited it already in other places?*

AR: I have added it

RC: *L503f: This is a very interesting thought. How many overpasses would you think it needs to assess that the emissions are stable enough to reduce revisiting times?*

AR: We need regular basin monitoring to answer this question. Ultimately we are interested in how total basin emissions are trending over time. The more variable each successive basin emissions estimate is, the more uncertain we are in the current mean emissions from the basin, implying the need for more frequent observation. Hypothetically, one could estimate the minimum sampling frequency by considering the current sample variance, and defining a minimum threshold trend that is needed to be detected over a specified time range. For a given sampling frequency one could then define a linear model over the time range, with the given minimum slope and data variance, and via a t-test determine if this would be statistically significant. As of now this is not in the target scheduling plan but it might be an interesting idea.

RC: *L509: Which of the nine overpasses do you consider covering the plume fully? In Fig. 19, I see six plumes I would consider "cut off" and only three plumes fully sampled.*

AR: Originally I considered RF06 Passes 2,3,4,5 abd RF07 2,3 - the swath slighty cuts off the enhancements above background in some of these, but their emission rates are reasonably consistent. The comment was made to explain that RF06 overpasses 1 and 7 have lower emissions estimates due to poor observation overlap. I have changed to the text clarify:

> The estimated emissions were persistent and large, ranging from 1914–2572 kg/h upon excluding RF06 16:15 and RF07 17:11 due to poor observational coverage of the target site.

RC: *L530: While true, such large persistent leaks have also been detected by TROPOMI, GHGSat, and EMIT, e.g., in Turkmenistan. Therefore, it enhances the capabilities to inform operators but is not the only instrument for that, which is implied by the statement.*

**AR:** Updated to:

> This case demonstrates that even large-magnitude leaks can go undetected for long periods of time, and suggests that through regular OG basin monitoring, MethaneSAT will provide an important addition to the existing network of satellites capable of detecting large methane leaks (Irakulis-Loitxate et al., 2022), and help contribute to their mitigation through international activities such as the Methane Alert and Response System (UNEP, 2023).

**RC:** *Figure 20: Was this plume recorded in one go, or has it been sampled over multiple tracks? If the latter is the case, over which time frame was the plume recorded, and how do you treat the wind in this case for the emission estimate?*

**AR:** The plume was sampled over 3 successive overpasses approximately perpendicular to the plume orientation, intersecting at approximately 15-20 minute intervals. The IME was derived from this data after it was combined and regridded at a resolution of $\sim 10 \times 10\mathrm{m}^2$. The effective wind speed was estimated from the 10-m RMS wind speed from an LES simulation at the time of the overpass corresponding to the pipeline leak location. In practice this effective wind will work better for smaller plumes - when the plume becomes larger the $U_{eff}$ may not be representative of that downwind, and additional internal eddy structures deep within the plume will not necessarily aid in its dissipation, causing the LES to underestimate the plume lifetime. We do note however that the confidence interval of the IME (5700-6500 kg/h) is within that derived using the divergence integral approach (5600-5700 kg/h, see Chulakadaba et al. (2023)) which should be less susceptible to errors caused by the larger spatial extent of the plume.

**RC:** *L533: "the operational MethaneSAT" – please add "designated" as MethaneSAT is not flying yet.*

**AR:** Changed to: "Here we have described the retrieval algorithm designated for MethaneSAT"

**RC:** *L545f: Does this assume purely random noise? What about systematic effects that cannot be reduced by averaging?*

**AR:** Yes we are referring to random error, hence why precision is used in the sentence.

**4. Technical Corrections/Typos**

**RC:** *L220: "clouds with high-optical": here a word seems to be missing*

**AR:** Corrected to "high optical depths"

**RC:** *L237: "... is the mean ...": "The" missing*

**AR:** corrected

**RC:** *L264: mechanistically sounds strange here – do you mean "directly"?*

**AR:** Both the change in ISRF width and CH4 bias are both emergent properties caused by the defocusing of the instrument. The term mechanistically was meant in the scientific sense, whereby a mechanism is understood as a system of "causally interacting parts and processes that produce one or more effects." To avoid confusion I have changed the sentence to

> Since there is an underlying physical connection between the XCH4 cross-track bias and ISRF squeeze parameters, . . .

**RC:** *L267: please replace "(top right") with the according subfigure number*

AR:  Top right $->$ c2 (was referencing an older version of figure)

**RC:** *L299: "2-3 hours of MethaneAIR observations" – please add at which altitude, as this determines the swath width.*

AR:  Added "at its nominal 12km above-ground observation altitude."

**RC:** *Figure 12: Typo in the caption: "wiht" -> with*

AR:  Fixed

**RC:** *Figure 14: Label and ticks of the y-axis on the right plot are covered*

AR:  Fixed

**RC:** *L511: There is no reference in the bibliography for the US EPA GHG reporting program, or it is not listed under this name.*

AR:  Citation to EPA GHGRP URL added (GHGRP)

**RC:** *L512: The study's measurements were taken ten years ago, and the paper was published eight years ago, so please remove the "recent".*

AR:  Done

**RC:** *L525: Please give the 2.7 Tg a-1 also in kg h-1 for better comparison with the other values in the paragraph.*

AR:  Typically basinwide totals as Gg/a or Tg/a, and plume sources in kg/h. Converting to kg/h yields 3.1e6 kg/h which seems a bit unwieldy. The purpose of the sentence was just to put into perspective the size of the source relative to the basin, which we also indicate as a percentage.

**RC:** *L543/544: two times "strong"*

AR:  Fixed

**References**

[revised manuscript text omitted]

---

## Author Comment (AC2)

**Authors' Response to Reviews of**

**Methane retrieval from MethaneAIR using the CO2 Proxy Approach: A demonstration for the upcoming MethaneSAT mission**

,
* * *
**RC:** *Reviewers' Comment*,     AR: Authors' Response,     ☐ Manuscript Text

**1. Comprehension**

**RC:** *The manuscript presents shortwave infrared methane measurements using MethaneAIR, an airborne demonstrator for the MethaneSAT satellite. The study focuses on retrieval performance and validation of the CO2 proxy method for CH4 retrieval, addressing instrumental challenges like ISRF drifts, and implementing bias correction techniques. It also validates MethaneAIR's measurements against ground-based data. The preprint aims to extrapolate its findings to anticipate the performance of the forthcoming MethaneSAT mission, particularly in terms of precision and detection limits for methane emissions. The study presents valuable and relevant research but necessitates revisions, particularly in the description of methodologies and structural organization, to ensure clarity and completeness before it is suitable for publication.*

We thank the reviewer for their time and helpful suggestions for the manuscript. Our responses to their comments are below.

**2. General Comments**

**RC:** *The manuscript's extrapolation of findings from MethaneAIR to MethaneSAT requires improvement, as it frequently lacks comprehensible detail on how this projection is methodologically executed (including the assumptions and limitations).*

**AR:** We have added a section (S2) to the supplement to address these issues. This includes performing a linear sensitivity analysis to estimate the measurement precision of MethaneSAT with what is currently known about the sensor (Section S2.1).

**RC:** *What types of errors can lower spatial resolution introduce, such as the omission of low-contrast pixels, among others?*

**AR:** Section S3 now includes additional details on potential issues caused by the lower spatial resolution of MethaneSAT. These include

- Cloud contamination (Section S2.3.1)

- Inhomgeneous illumination (Section S2.3.2)

- Sub-pixel $CH_4$ concentration gradients (Section S2.3.3)

In general we do not find significant differences in performance between the two sensors for the error sources considered.

**RC:** *The discussion about e.g. the 0.15% target precision for MethaneSAT and its relation to MethaneAIR's performance is limited and I can't clearly see how this extrapolation is methodologically done.*

AR: This was just based on the central limit theoerem. By aggregating pixels the precision of the sample mean concentration ($\bar{\sigma}$) should follow

$$\bar{\sigma} = \frac{\sigma_0}{\sqrt{n}} \tag{1}$$

Where $n$ is the number of pixels and $\sigma_0$ is the single-pixel precision. We have added a footnote to clarify how this was calculated.

**RC:** *A numerical analysis of the uncertainties associated with emission estimates would be desirable.*

We have updated the manuscript to include emissions uncertainties for the emissions estimates derived from the IME method. The methodology for these estimates is described in the manuscript cited in the paper (Chulakadaba et al., 2023).

**RC:** *Section 8 of the manuscript is relatively brief considering its title, suggesting a need for more extensive coverage and depth in its content.*

AR: We have expanded the MethaneSAT discussion, with links to the supplement for further information

> Importantly, we find no strong dependence on surface reflectance in the MethaneAIR results. This was a key untested assumption in the early emissions inversions observing system simulation experiments that were used to inform the MethaneSAT instrument requirements (Benmergui, 2019). In those original experiments the assumed precision of MethaneSAT was 0.15% at $1 \times 1$ km$^2$ resolution. In follow up experiments it was found that the satellite could meet its emission constraint goals with that precision at a scale of $5 \times 5$ km$^2$. We expect MethaneSAT to have a similar per-pixel precision as MethaneAIR (35 ppb, Section S2.1). At these levels the 0.15% target precision will be achieved at a scale of $\sim 3 \times 3$ km$^{2a}$, well within the precision requirement. Although the instruments are at different spatial resolution, we do not find an large differences in cloud contamination (Section S2.3.1), or biases caused by sub-pixel inhomogeneities in illumination (Section S2.3.2) and methane concentration (Section S2.3.3) between the two instruments.
>
> ---
> [a]Calculation based on central limit theorem. 0.15% translates to 2.85 ppb assuming a 1900 ppb $XCH_4$. The precision of the sample mean reaches the requirement for $\sim 150$ pixels, which translates to $\sim 3 \times 3$ km$^2$, assuming a $140 \times 400$ m$^2$ native pixel size.

**RC:** *Certain sections of the paper could benefit from a more concise description, and the overall structure could be improved for clarity and better organization. Consider introducing additional subsections to achieve this.*

AR: We have implemented the majority of subtitle suggestions by the first reviewer, which should cover this point.

**RC:** *Consider to display the mathematical expression used to compute the squeeze factor in the forward model. Overall, provide some more details on the forward model (e.g. input quantities, etc.).*

AR: We have significantly expanded Section 1 of the supplement. It now contains a comprehensive description of the forward model.

RC: *Emphasize that the ISRF squeeze is a nominal fit parameter and clarify that the remaining bias is a result of defocusing, which cannot be resolved solely through the ISRF squeeze adjustment but necessitates an additional preprocessing step, such as PLS regression.*

AR: It already does say this in Section 4, explaining the need for the cross track regression.

> It is unlikely that squeezing the tabulated laboratory ISRF fully accounts for the change in ISRF shape induced by defocusing. The gradual drift in instrument focus may lead to a time-dependent $XCH_4$ cross track bias, which we attempt to derive using a "small area approximation" (O'Dell et al., 2018).

RC: *The bias analysis would benefit from additional details regarding the underlying assumptions and background information.*

The additional changes recommended by reviewer 1 for clarification in the section hopefully address this comment.

**3. Specific Comments**

RC: *63: Consider to add some recent literature sources on airborne methane retrieval (e.g. methane by HySpex)*

AR: We have added HySpex to the list of airborne sensors already cited

> Recently instruments designed to detect high concentrations of $CH_4$ in individual methane plumes have been deployed on aircraft (AVIRIS-Thorpe et al. (2012), AVIRIS-NG-Thorpe et al. (2016), HySpex-Hochstaffl et al. (2023)) and satellites (Sentinel 2-Varon et al. (2021), GHGSat-Jervis et al. (2021), CarbonMapper-Shivers et al. (2021), PRISMA-Guanter et al. (2021), EnMAP-Roger et al. (2024))

And MAMAP later:

> The $CO_2$-Proxy method was first used from an airborne platform by the MAMAP instrument (Krings et al., 2011; Gerilowski et al., 2011), ...

RC: *99: Could you provide some brief insights into the command chain involved in operations like near realtime satellite commanding, which relies on forecasts and other factors? I*

AR: I have added a citation to the PhD thesis that provides the details of the scheduling system (Benmergui, 2019).

RC: *132: The accuracy of the proxy method could potentially be compromised if there is gas flaring occurring in close proximity?*

We have updated the supplement (Section S2.2) to assess errors caused by $CO_2$ co-emission from gas flares. We estimate based on currently known distributions of gas flaring efficiencies that flare emissions from US oil and gas basins will be underestimated by 8-16% depending on the basin. We are currently assessing alternate approaches for situations where we expect local CO2 enhancements including using the unnormalized columns, or proxy-normalization from the O2 singlet-delta band.

**RC:** *137: The MethaneAIR ground pixel size significantly differs from the nominal resolution of MethaneSAT. Any conclusions for MethaneSAT need to address assumptions and limitations in such a projection.*

**AR:** This is now addressed in Section 2.3 of the supplement, as already discussed in the general comments.

**RC:** *185: Consider to be more explicit. But if I understand correctly, there are a total of 19 layers from the Bottom of the Atmosphere (BOA) to the Top of the Atmosphere (TOA), with 13 of these layers situated within the troposphere. 18 + Table 1: Why scale all 19 layers when the enhancement primarily originates at or near the surface and likely remains concentrated in the lower layers for several kilometers, especially under steady wind and stable atmospheric conditions?*

**AR:** The enhancement above background occurs in the lowest layers, however for an individual sounding there is also the bias in the a priori background. The observed CH4 line absorption depths are larger for the same increase in CH4 molecular density for lower layers due to the effect of pressure broadening. Thus if the background was under/overestimated by the prior, scaling only the surface layer will under/overestimate the column XCH4. The reverse is true if we scaled the entire column.

We currently optimize the 19 levels, but they are constrained by a prior covariance matrix (figure S1) that specifies correlations between the levels. This is the approach used by other satellite proxy retrieval algorithms (e.g. SCIAMACHY - Frankenberg et al. (2006), GOSAT - Parker et al. (2020)). Here we have adjusted the prior covariance such that the response to boundary layer differences is close to 1:1 (Averaging kernels, Figure S2).

**RC:** *199: An Instrument Spectral Response Function (ISRF) squeeze factor of less than 1 typically indicates squeezing, while a factor greater than 1 signifies stretching. So, ISRF squeeze < 1 means squeezing, and ISRF squeeze > 1 means stretching? Be more explicit.*

**AR:** Added sentence after equation:

> From Equation 3, $x_{sqz}$ values below/above unity correspond to stretching/squeezing the tabulated ISRF respectively. "

**RC:** *203: Where is this shown? Eq.(3): Consider to include the complete formula for the squeeze factor.*

**AR:** Figure 3 - middle panel - I have put it in brackets after the statement. Eq. 3 is the complete formula for the squeeze factor.

**RC:** *220: "... high optical depth,..."? Fig. 4: Is it plausible or reasonable to assume that XCH4 (XCH4 typically refers to column-averaged dry-air mole fraction of methane) is enhanced at the position of the cloud?*

**AR:** Yes it's a typo for high optical depth - now fixed

The paragraph refers to "CH4 vertical column densities (VCDs)" not XCH4 i.e. the vertically integrated number density (molecules/unit area). High optical depth clouds will in most cases have a light path shortening effect, leading to a decrease in CH4 VCD relative to the value computed from the a priori. This is in essence the principle used by the OCO2 A-band cloud processor. The only difference is that there is a higher uncertainty in the CH4 relative to the prior. We estimate the background uncertainty using the data pre-screened using data from the O2 band (where it spatially overlaps), and set this as the baseline to account for a priori CH4 VCD uncertainties. We estimate a probability of a cloud based on the distance the retrieved CH4 VCD is away from this baseline. We do the same for CO2 and combine the values from all 3 (O2, CH4, CO2) using a naive Bayes classifier. The full method will be available in a forthcoming publication.

**RC:** *264: Is "mechanistically" the right term?*

**AR:** See comment to Reviewer 1. Changed sentence to:

> Since there is an underlying physical connection between the XCH4 cross-track bias and ISRF squeeze parameters, ...

**RC:** *268: "… 10 s of observations"? Fig. 5: In (a1) and (a2), where is the representation of the XCH4 bias located or depicted?*

**AR:** It should be Fig. 5 (c2) - text updated

> As the noise in retrieved ISRF squeeze parameters is lower than the retrieved $XCH_4$, this will also improve the precision of cross-track bias prediction compared to direct application of the values in Figure 5 (c2).

**RC:** *276: Repeating "valuable"*

**AR:** Fixed

**RC:** *Fig. 7: Check first sentence in caption. Add XCH4 label to colorbar. Show Beta from Eq. (5) or at least add some information in the caption that helps to better relate the figure with Eq. (5).*

**AR:** Added "$XCH_4$ Bias" label to colorbar.

I have updated the caption in Figure 7:

> $XCH4$ cross track bias estimate with the small area approximation (left) and subsequently refined using the PLS regression model (right). The PLS regression model (Equation 5) uses the retrieved ISRF squeeze factors to predict the data in the left panel, in order to preserve only sources of variability related to the temperature-induced defocusing effects.

**RC:** *318: Clarify.*

**AR:** Smaller particles lead to larger angstrom exponents, suggesting that even though smoke could be seen at visible wavelengths, it did not scatter sufficiently in the SWIR where the observations were made.

**RC:** *341: Was the TROPOMI L2 product destriped as part of the processing, or was it delivered already destriped? My assumption is that they provide a destriped product.*

**AR:** They do not (or at least did not at the time of submission). I speculate it is an operational processing scheduling issue. If a product requires $n$ previous days of observations to make the correction, then this means that production of the final version of that file would be delayed by that amount of time.

**RC:** *369-372: The XCH4 albedo dependence analysis in Fig. 12 is conducted only for a single scene. Is this sentence required?*

**AR:** The scene is representative of data from the campaign. The low sensitivity to surface features is important to emphasize, as it is critical for enabling quantification of methane emissions below the plume detection limit. This is an overarching goal of the MethaneSAT/MethaneAIR project.

**RC:** *377: Where to find Fig. S3?*

AR:   In the supplement - I think this is standard notation for this, but open to suggestions.

RC:   *401: Review the sentence (IME is not directly a plume detection method).*

AR:   Changed to:

> The filter was chosen for its favorable properties, such that the resulting smoothed XCH4 fields could be used to estimate $CH_4$ point source emissions ($Q$) without inducing additional bias. Here and for MethaneSAT, the integrated mass enhancement (IME) method (Frankenberg et al., 2016; Varon et al., 2018) is the primary method used to estimate $Q$:

RC:   *Fig. 8: Add more details to the caption. The stripes are in the along-track direction due to the cross-track bias? Do I see multiple parallel flight tracks here?*

AR:   Added context to flight path:

> Here the aircraft is traveling in a clockwise loop, with the northern segments overlapping to target a controlled release.

RC:   *Fig 9  Fig. 10: Was the averaging kernel actually considered in the comparison? It may be beneficial to show the formula to accommodate the varying vertical.*

AR:   The averaging kernel can be used to adjust both sensors to a single prior - however in this case they are both using almost identical GGG2020 profiles (the only difference is that they are calculated with GEOS-FP vs. GEOS-FPIT met products, the latter being a frozen GEOS-FP release to maintain consistency for long-term datasets). If there was a coincident in situ column measurement then both could be compared to that, but we do not have this.

The other alternative is to estimate the distribution of expected differences using an ensemble estimate to anchor both soundings to - see Rodgers and Connor (2003), section 4.3. In this case you still need to define the probability distribution of the ensemble. Another approach for accounting for the vertical sensitivity difference would be to smooth the retrieved EM27 profiles by the MethaneAIR averaging kernels. However this will not factor in errors in the EM27 averaging kernel, and since its stratospheric values are changing significantly between some of the overpasses we thought it easier to just discuss the associated smoothing errors in the context of GGG2020 uncertainties. If the reviewer believes this is worthwhile, we can update the figure with this additional method.

RC:   *Fig 12: Consider providing a brief description of the y-axis label in the caption. Additionally, could you explain why the lines do not exhibit a consistent trend towards either higher or lower values? If scattering is the underlying process responsible for this bias, it's puzzling why it does not consistently manifest as either a positive bias (indicative of multiple scattering) or a negative bias (indicative of single scattering).*

AR:   The description is in the caption - updated to explicitly call it that

> $XCH_4$ bias (y-axis) is derived by subtracting the mean $XCH_4$ for albedos between 0.19–0.21.

The discussion of the trend is already in the discussion of the figure in Section 5.3 - the reason for the slight upward trend at lower albedos is regularization bias (L375 in discussion paper). The bias for high albedos is due to a few unfiltered clouds. This is why there is not a consistent sign, as it is not one phenomenon.

Our main point was that the albedo-induced bias is less for the proxy-approach compared to other XCH4 retrievals (either low res spectrometers like AVIRIS whereby the spectral information is difficult to separate from albedo) or strong CH4 band full-physics retrievals like TROPOMI, that require aerosol optical property assumptions to connect the CH4 estimate to the spectrally separated light-path constraint.

**RC:** *390: Please provide details on how the value of 35 ppb was determined. Is it based on the standard deviation or another statistical measure? Fig 14: Check axis labels on right panel.*

AR: It is the precision determined by the standard deviation of the measurements in Fig 12. The manuscript has been updated to make this explicit

> We estimate the precision of the $5 \times 1$ aggregated retrievals of 35 ppb, by taking the the standard deviation of the $XCH_4$ retrieved over background locations used in Figure 12.

The Axis label on the right panel is now fixed.

**RC:** *545: The resolution is similar to TROPOMI's resolution (at least it is in the same order of magnitude.)*

AR: The 0.15% precision refers to the target precision at a scale of 3x3 km$^2$. The native MethaneSAT pixel is 400x130m$^2$ (change subject to final orbit altitude). In contrast, TROPOMI's precision for a single $5.5 \times 7$ km$^2$ pixel is 0.8%. The point of the sentence was to say what the threshold spatial scale is for detecting small XCH4 gradients (above 3 ppb).

**RC:** *545: Provide more context or specify where the analysis yielding the result of 0.15% is located? Additionally, it would be helpful to understand the assumptions made to arrive at this conclusion.*

AR: We have updated the paragraph to provide the context for the 0.15% target threshold, which was based on emission inversion observing system simulation experiments. The way the instrument precision was reached was based on precision estimates from a linear sensitivity analysis (Section S2.1). The estimate that the precision will be achieved at the $3 \times 3$ km$^2$ spatial scale just follows from the central limit theorem, as discussed earlier in these responses.

**References**

[revised manuscript text omitted]

---

## Referee Report (RR1)

**Review of the revised manuscript**

*"Methane retrieval from MethaneAIR using the CO2 Proxy Approach: A demonstration for the upcoming MethaneSAT mission" by Chan Miller et al.*

**General comments**

The authors have added references and clarified methodologies as requested and expanded sections in the supplement to address the feedback on detailing the mathematical bases and assumptions of their model, which I previously considered insufficient in some parts. They made significant updates where needed, while also justifying their original approaches when they disagreed with the comments.

Overall, their response improves the manuscript's clarity and demonstrates a commitment to addressing the reviewer's concerns. Therefore, the manuscript should be published after minor revisions are made.

**Specific comments**

*Note that my line numbers refer to the version of the document that includes red and blue differences.*

Consider revising the order of the conclusions in Section 8 to present the most important aspects first, followed by the less critical ones. In my opinion the temperature-induced shifts, for instance, should not be the first aspect discussed in the conclusions.

Add clarification to distinguish between the terms "quantification" and "detection", noting the term "quantification" is italicized at line 589 and that the detection limit of 121 kg/h and the quantification limit of 200 kg/h are considered consistent.

It would be beneficial to include a table comparing the specifications of MethaneAIR and MethaneSAT, as numerous figures are mentioned throughout the manuscript.

15: Please ensure that the number 2.5 ppb for the latitude gradient bias is clearly referenced in the main text, as it currently appears to be missing from Section 5.2.

26: Consider to clarify what is meant by "fine spatial resolution" and "large swaths" in the context of MethaneAIR, and provide specific information similar to the details given for MethaneSAT in the following sentence.

42: Including a table comparing the specifications of MethaneAIR to MethaneSAT at this point would be helpful.

61: Consider to mention that scattering effects are more pronounced at shorter wavelengths, thus the 1.6 μm band is more affected than the 2.3 μm band in TROPOMI.

64: You mention data from aircraft but also refer to satellite platforms.

65: Does this relate even more to the precision requirement?

94: How do MethaneSAT's 20-30 revisits per year impact its claim or capability for monitoring?

104: Am I right that strictly speaking, it is the primary retrieval method used to infer concentration enhancements (which are then used in the emission inversion).

109: Review the phrase "sensor rate spatial coverage."

Fig. 2: Consider specifying what constitutes a "typical measurement" in terms of photon radiance or another metric.

165: Clarify what is meant by "an additional set of weaker lines overlap with the C."

231: Does the initial estimation of surface albedo consider only a single pixel?

Sec. 4.1: I suggest making the section more concise.

Fig. 5: Label the squeeze factor next to the colorbar in (c1), similar to how ppb is labeled in (c2). What cross-track index is displayed in (c2)?

274-279: This passage could be rewritten for better clarity.

393: The reference in Fig 11 to XCH4 being constant within 50 km is not entirely convincing (though it remains reasonable).

394: Consider the potential issues of relying on data from a single pixel, even if it's only for an a priori estimate.

395, Fig. 12: Would using the median be a more robust measure than the mean?

Fig. 12: Consider adding a colorbar.

395: Ensure consistency; the text mentions computing the mean from a 0.02 width bin, but the figure caption refers to ±0.1 width, while the gray lines represent 0.02 binned averages.

Fig. 13: Please describe the units of the colorbars.

426: Given the importance of the 35 ppb finding, consider adding a statement relating this to Section 2.1 of the attachment. Also, think about moving the precision finding of 35 ppb to Section 5.3, as it is based on data presented in that section.

430: How is the smoothed image 'g' calculated?

Fig. 15: Consider to ensure that the colorbars have specified lower and upper bounds. Also, consider specifying time in hh:mm:ss format as done in Fig. 17.

514-515: Clarify why the detection limit of 121 kg/h and the quantification limit of 200 kg/h are considered consistent.

496: This point suggests a broader issue regarding the estimation of average emission rates from irregular revisits. Consider adding a brief discussion on this topic in the discussion section.

500: Provide at least one argument or reference to support this statement.

Fig. 19: Discuss any implications for the IME method if the swath cuts the plumes. Adjust the axis labels and colorbar font sizes for better readability.

Fig. 20: Increase the colorbar font size.

S1.2: Briefly explain why the a priori profile pressure and temperature levels are dependent on the a priori surface and tropopause pressure, and how these coefficients are determined.

Eq. (S9): Describe how the optical depths are adjusted for an adjusted temperature.

Eq. (S8): Shouldn't the ratio yield a sigma coordinate on the left-hand side?

S2.3.2: Clarify whether the RR method is only feasible for aggregated pixels since single pixels cannot be divided into upper and lower halves. Explain the rationale for choosing upper/lower division over left/right.

S2.3.2, Fig. S10: Consider using colorbar annotations instead of titles for consistency.

Fig. S12: Confirm whether "Pixel Mean XCH4 Enhancement" is the correct label for the colorbar. Should it instead indicate enhancement in the plume, affecting the error depending on the plume's prominence?

---

## Author Response (AR2)

**Authors' Response to Reviews of**

**Methane retrieval from MethaneAIR using the CO2 Proxy Approach: A demonstration for the upcoming MethaneSAT mission**

,
* * *
RC: *Reviewers' Comment*,     AR: Authors' Response,     ☐ Manuscript Text

**1.  General Comments**

The authors have added references and clarified methodologies as requested and expanded sections in the supplement to address the feedback on detailing the mathematical bases and assumptions of their model, which I previously considered insufficient in some parts. They made significant updates where needed, while also justifying their original approaches when they disagreed with the comments.

Overall, their response improves the manuscript's clarity and demonstrates a commitment to addressing the reviewer's concerns. Therefore, the manuscript should be published after minor revisions are made.

**2.  Specific Comments**

RC:  *Consider revising the order of the conclusions in Section 8 to present the most important aspects first, followed by the less critical ones. In my opinion the temperature-induced shifts, for instance, should not be the first aspect discussed in the conclusions.*

AR:  As this is a style request, we respectively decline, as we prefer to list the conclusions in chronological order.

RC:  *Add clarification to distinguish between the terms "quantification" and "detection", noting the term "quantification" is italicized at line 589 and that the detection limit of 121 kg/h and the quantification limit of 200 kg/h are considered consistent.*

AR:  The term was italicized to emphasize that they are different metrics, as there was confusion in the preprint. I think the terms are self explanatory. Detection means that we can identify a source of methane, but the enhancement is too low for it to be quantified. Quantification means that it is large enough that it can be. Thus the detection limit should always be lower than the quantification limit.

RC:  *It would be beneficial to include a table comparing the specifications of MethaneAIR and MethaneSAT, as numerous figures are mentioned throughout the manuscript.*

AR:  Addressed in specific comment Line 42.

RC:  *15: Please ensure that the number 2.5 ppb for the latitude gradient bias is clearly referenced in the main text, as it currently appears to be missing from Section 5.2.*

AR:  The 2.5 ppb refers to the mean bias between the instruments. We have reworded the sentence to make this

clearer

> MethaneAIR retrievals were also intercompared with those of TROPOMI; The mean bias between instruments is 2.5 ppb, and the latitudinal gradients for the two datasets are in good agreement.

**RC:** *26: Consider to clarify what is meant by "fine spatial resolution" and "large swaths" in the context of MethaneAIR, and provide specific information similar to the details given for MethaneSAT in the following sentence.*

**AR:** We have clarified the sentence by attaching numbers to the spatial resolution and mapping area, and included the approximate target size for MethaneSAT.

> The results showcase the capability of MethaneAIR to make highly accurate, precise measurements of methane dry-air mole fractions in the atmosphere, with fine spatial resolution ($\sim 20 \times 20$ m$^2$) mapped over large swaths ($\sim 100 \times 100$ km$^2$) in a single flight. The results provide confidence that MethaneSAT can make such measurements at unprecedentedly fine scales from space ($\sim 130 \times 400$ m$^2$ pixel size over $\sim 200 \times 200$ km$^2$ target area), thereby delivering quantitative data on basin-wide methane emissions.

**RC:** *42: Including a table comparing the specifications of MethaneAIR to MethaneSAT at this point would be helpful.*

**AR:** This can be found in Chulakadaba et al. (2023). We cite this with reference to the table in the updated manuscript

> In preparation for MethaneSAT's launch, an airborne precursor called MethaneAIR has been constructed (Staebell et al., 2021), with near-identical instrument specifications (Table S1, Chulakadaba et al. (2023)).

**RC:** *61: Consider to mention that scattering effects are more pronounced at shorter wavelengths, thus the 1.6 m band is more affected than the 2.3 m band in TROPOMI.*

**AR:** Whilst this is true, we mention the target band only to state that the proxy method is not possible. The main point of the paragraph was that the proxy method generally has a higher per-pixel retrieval success rate.

**RC:** *64: You mention data from aircraft but also refer to satellite platforms.*

**AR:** The sentence does already distinguish between the aircraft and satellite based instruments (bold for emphasis)

> Recently instruments designed to detect high concentrations of CH$_4$ in individual methane plumes have been deployed **on aircraft**(AVIRIS-Thorpe et al. (2012), AVIRIS-NG-Thorpe et al. (2016), HySpex-Hochstaffl et al. (2023)) **and satellites**(Sentinel 2-Varon et al. (2021), GHGSat-Jervis et al. (2021), CarbonMapper-Shivers et al. (2021), PRISMA-Guanter et al. (2021), EnMAP-Roger et al. (2024)) to estimate emission rates from point sources.

**RC:** *65: Does this relate even more to the precision requirement?*

**AR:** The AVIRIS like sensors typically have large biases (10-100 ppb XCH4) that strongly correlated to surface, as at such coarse spectral resolutions the surface cannot be easily disentangled from methane absorption.

At spatial scales beyond 1-10m, such biases would remain, by most sources would produce enhancements smaller than them. This is why we emphasize that it enables loosening the accuracy requirement.

**RC:**   *94: How do MethaneSAT's 20-30 revisits per year impact its claim or capability for monitoring?*

AR:   Whilst MethaneSAT will be able to contribute to anomalous event reporting by partnering with organizations such as the UN IMEO program, a chief aim of the mission is to determine how methane emissions from each major production basin evolve over time (e.g. as verification that producers are meeting their COP pledges). The planned 10-20 revisits per year should be enough to detect changes associated with changes from improvements in infrastructure and production practices.

**RC:**   *104: Am I right that strictly speaking, it is the primary retrieval method used to infer concentration enhancements (which are then used in the emission inversion).*

AR:   Yes - this is now clarified in the updated manuscript

> Here we present results from the maiden flight campaign of MethaneAIR using the operational MethaneSAT $CO_2$-proxy $XCH_4$ retrieval. This is expected to be the primary $XCH_4$ product used in subsequent emissions inversions

**RC:**   *109: Review the phrase "sensor rate spatial coverage."*

AR:   Fixed - it should be "sensor's spatial coverage rate"

**RC:**   *Fig. 2: Consider specifying what constitutes a "typical measurement" in terms of photon radiance or another metric.*

AR:   The manuscript has been updated to reference the albedo used in the simulation.

> Example transmission spectra for a typical MethaneAIR observation (0.3 Lambertian albedo, 30° solar zenith angle).

**RC:**   *165: Clarify what is meant by "an additional set of weaker lines overlap with the C."*

AR:   There are more CO2 lines in the same wavelength range covered by the R branch of the CH4 band. I'm not sure how to write this more clearly.

**RC:**   *231: Does the initial estimation of surface albedo consider only a single pixel?*

AR:   It is done for each pixel. The manuscript has been updated to clarify this.

> The *a priori* Lambertian surface albedo for each pixel is computed using the transparent region of the observed radiance at 1622 nm, assuming a non-scattering atmosphere.

**RC:**   *Sec. 4.1: I suggest making the section more concise.*

AR:   The discovery and correction of the instrument defocusing induced by the unstable temperature environment was a big retrieval challenge. The section is long because the origin had to be investigated, and the impact on $XCH_4$ bias carefully characterized.

**RC:**   *Fig. 5: Label the squeeze factor next to the colorbar in (c1), similar to how ppb is labeled in (c2). What cross-track index is displayed in (c2)?*

**AR:** I have updated the colorbar label in c1 ($x_{sqz}$) I am not sure what is meant by the second sentence. The x-axis corresponds to the cross-track index of the detector.

**RC:** *274-279: This passage could be rewritten for better clarity.*

**AR:** The passage was written as a suggestion by reviewer 1.

**RC:** *393: The reference in Fig 11 to XCH4 being constant within 50 km is not entirely convincing (though it remains reasonable).*

**AR:** Provided that there is a similar albedo variation across the 3-min segment the XCH4-albedo anomaly should still be detectable. The TROPOMI analysis from which the figure is based was performed over much larger geographic areas ( 5 degrees latitude).

**RC:** *394: Consider the potential issues of relying on data from a single pixel, even if it's only for an a priori estimate.*

**AR:** The value is the binned mean (red line) for the mean of all pixels within the 0.19-0.21 albedo bin (the red line in Figure 12). The 2-sigma sample mean uncertainties (dashed red lines) are also shown in the figure. The uncertainty is negligible.

**RC:** *395, Fig. 12: Would using the median be a more robust measure than the mean?*

**AR:** In general yes, but Fig. 12 also shows the data density - there appears to be no outliers. Thus the mean and median are quite similar.

**RC:** *Fig. 12: Consider adding a colorbar.*

**AR:** A colorbar has been added to the figure in the updated manuscript

**RC:** *395: Ensure consistency; the text mentions computing the mean from a 0.02 width bin, but the figure caption refers to 0.1 width, while the gray lines represent 0.02 binned averages.*

**AR:** The text says a 0.02 width bin centered at 0.2, and the figure says the bin covers the range 0.19-0.21. This is the same. The text is not referring to the grey lines, but the specific bin at 0.2.

The red and grey lines correspond to all the binned averages. The text is updated to say this more clearly.

> The grey and red lines show the binned averages computed in 0.02 albedo increments before and after cloud screening respectively

**RC:** *Fig. 13: Please describe the units of the colorbars.*

**AR:** The updated manuscript includes labels for the colorbars in the figure

**RC:** *426: Given the importance of the 35 ppb finding, consider adding a statement relating this to Section 2.1 of the attachment. Also, think about moving the precision finding of 35 ppb to Section 5.3, as it is based on data presented in that section.*

**AR:** We have updated the manuscript to refer to Section S2.1

> The previous section showed that the main error in the flight retrievals is random noise. We estimate the precision of the $5 \times 1$ aggregated retrievals of 35 ppb, by taking the standard deviation of the $XCH_4$

retrieved over background locations used in Figure **??**. This value is consistent with our estimate for the native resolution of MethaneSAT (Section S2.1), which has similar SNR to the $5 \times 1$ aggregated MethaneAIR retrieval. These noise levels reduce MethaneAIR's ability to detect small-scale $XCH_4$ gradients.

**RC:** *430: How is the smoothed image 'g' calculated?*

AR: By finding the $g$ that minimizes equation 8, as written in the text.

**RC:** *Fig. 15: Consider to ensure that the colorbars have specified lower and upper bounds. Also, consider specifying time in hh:mm:ss format as done in Fig. 17.*

AR: We have updated the figure based on the reviewer suggestions

**RC:** *514-515: Clarify why the detection limit of 121 kg/h and the quantification limit of 200 kg/h are considered consistent.*

AR: It is similar in magnitude but lower than the quantification limit. That is all that is meant by that statement.

**RC:** *496: This point suggests a broader issue regarding the estimation of average emission rates from irregular revisits. Consider adding a brief discussion on this topic in the discussion section.*

AR: The subject of source intermittency on day-by-day timescales is discussed elsewhere in the literature (e.g. Cusworth et al. (2021)). The point we are trying to make is that since are detecting disconnected methane plumes, this will cause complications for the emissions inversion, whereby any source of methane is considered to be emitting at a constant rate over the course of the observation period. It is an unintended consequence of having accurate $XCH_4$ data at fine spatial scales, and must be considered by the emissions inversion model.

**RC:** *500: Provide at least one argument or reference to support this statement.*

AR: This is based on the scale of the disconnected plume O(1 km$^2$), the size of the enhancement (O(10 ppb)) and the spatial resolution (O(10 $\times$ 10 m$^2$) and O(100 $\times$ 100 m$^2$))/precision (35/30 ppb per pixel) of both instruments. At the scale of the enhancement there are O($10^4$) and O($10^2$) pixels, which means the precision over the enhancements is  0.03 and and 3 ppb respectively, well below the size of the disconnected plume in both cases.

We have updated the line

> We see here that they are observable by MethaneAIR and should be detectable by MethaneSAT, based on the size of the observed enhancement and the spatial resolution and precision of both sensors.

**RC:** *Fig. 19: Discuss any implications for the IME method if the swath cuts the plumes. Adjust the axis labels and colorbar font sizes for better readability.*

AR: The figure caption has been updated to indicate that partially-observed plumes will have their emissions underestimated.

> The IME method currently does not account for the impact of partially observed plumes; Such cases will lead to emission underestimates.

The size of colorbar/axes labels in Fig. 19 have been increased in the updated manuscript.

**RC:** *Fig. 20: Increase the colorbar font size.*

AR:  The colorbar font size has been increased in the updated manuscript

RC:  ***S1.2: Briefly explain why the a priori profile pressure and temperature levels are dependent on the a priori surface and tropopause pressure, and how these coefficients are determined.***

AR:  The coefficients are from the same vertical grid parameterization used by the University of Leicester GOSAT $CO_2$ retrieval. They are chosen to be compatible with the corresponding $CO_2$ and $CH_4$ a priori covariance matrices. We have updated the supplement as follows:

> The coefficients $a_l$, $b_l$, and $c_l$ are those used by the University of Leicester GOSAT $CO_2$-Proxy retrieval (Parker et al., 2020), and are provided below:

For the retrieval there is an advantage of including the tropopause pressure in the vertical grid parameterization because the stratospheric/tropospheric concentrations are not expected to correlate with one another due to the slow vertical transport timescale.

RC:  ***Eq. (S9): Describe how the optical depths are adjusted for an adjusted temperature.***

The text after equation 9 is updated to explain this

> The main effect of temperature is changing the absorption cross sections. In the retrieval the absorption cross sections are re-interpolated from their lookup tables each iteration to account for changes from the pressure/temperature state vector elements.

RC:  ***Eq. (S8): Shouldn't the ratio yield a sigma coordinate on the left-hand side?***

Although the initial pressure profile is defined with the surface/tropopause pressure hybrid grid, the pressure is optimized using sigma coordinates. This is already explained in the supplement immediately before the equation

> Whilst the initial pressure grid uses the hybrid parameterization in the previous section, surface pressures are optimized using sigma coordinates.

RC:  ***S2.3.2: Clarify whether the RR method is only feasible for aggregated pixels since single pixels cannot be divided into upper and lower halves. Explain the rationale for choosing upper/lower division over left/right.***

As explained in the text, the RR for the along-track half-pixels was estimated by linear interpolation between the observed albedos, assumed to be at the pixel centers.

> Since the scene is at the retrieved resolution, we have estimated $RR$ via linearly interpolating the albedos between pixels in the along track direction.

We argue that because this strongly correlates to the actual retrieved wavelength shifts, that this is a reasonable metric for assessing the degree of inhomogeneous illumination. The reason for choosing the along-track direction is that this is the direction that impacts the ISRF for our instrument. For more theoretical details, Appendix by of Landgraf et al. (2016) applies here. I have added a line indicating why we make this choice

> The inhomogeneity can be quantified by dividing each pixel in the along-track direction into a lower and upper half (indexed $l$ and $u$ respectively). The pixel is divided in the along-track direction because this is the direction that strongly impacts the ISRF.

**RC:** *S2.3.2, Fig. S10: Consider using colorbar annotations instead of titles for consistency.*

**AR:** The figure in the updated manuscript now has the labels as suggested

**RC:** *Fig. S12: Confirm whether "Pixel Mean XCH4 Enhancement" is the correct label for the colorbar. Should it instead indicate enhancement in the plume, affecting the error depending on the plume's prominence?*

**AR:** The figures actually correspond to the pixel-mean enhancement. e.g. if pixel mean enhancement is 100 ppb and the plume area fraction is 50%, this means the plume concentration will be 200 ppb. Each line corresponds to the same overall concentration enhancement - the x-axis indicates how it is distributed within the pixel.

**References**

[revised manuscript text omitted]